# Transcription-induced formation of extrachromosomal DNA during yeast ageing

Ryan M. Hull[1¤a], Michelle King[1], Grazia Pizza[1¤b], Felix Krueger[2], Xabier Vergara[1¤c], Jonathan Houseley[1]*

1 Epigenetics Programme, Babraham Institute, Cambridge, United Kingdom, 2 Babraham Bioinformatics, Babraham Institute, Cambridge, United Kingdom

¤a Current address: SciLifeLab, Department of Microbiology, Tumor and Cell Biology, Karolinska Institutet, Solna, Sweden.
¤b Current address: Section of Cell Biology and Functional Genomics, Division of Diabetes, Endocrinology & Metabolism, Department of Medicine, Imperial College London, London, United Kingdom
¤c Current address: Cell Biology department and Gene regulation department, Netherlands Cancer Institute, Amsterdam, Netherlands
* jon.houseley@babraham.ac.uk

**Data Availability Statement:** All relevant data are within the paper and its Supporting Information files. All sequencing files are available from the GEO database (accession number GSE135542).

## Abstract

Extrachromosomal circular DNA (eccDNA) facilitates adaptive evolution by allowing rapid and extensive gene copy number variation and is implicated in the pathology of cancer and ageing. Here, we demonstrate that yeast aged under environmental copper accumulate high levels of eccDNA containing the copper-resistance gene *CUP1*. Transcription of the tandemly repeated *CUP1* gene causes *CUP1* eccDNA accumulation, which occurs in the absence of phenotypic selection. We have developed a sensitive and quantitative eccDNA sequencing pipeline that reveals *CUP1* eccDNA accumulation on copper exposure to be exquisitely site specific, with no other detectable changes across the eccDNA complement. eccDNA forms de novo from the *CUP1* locus through processing of DNA double-strand breaks (DSBs) by Sae2, Mre11 and Mus81, and genome-wide analyses show that other protein coding eccDNA species in aged yeast share a similar biogenesis pathway. Although abundant, we find that *CUP1* eccDNA does not replicate efficiently, and high-copy numbers in aged cells arise through frequent formation events combined with asymmetric DNA segregation. The transcriptional stimulation of *CUP1* eccDNA formation shows that age-linked genetic change varies with transcription pattern, resulting in gene copy number profiles tailored by environment.

## Introduction

In contrast to the normally sedate evolution of chromosomal DNA, extrachromosomal circular DNA (eccDNA) can be rapidly accumulated and lost in eukaryotic cells, facilitating timely changes in gene expression and accelerating adaptation. eccDNA accumulation provides a pathway for adapting to drug treatment, environmental stress, and genetic deficiency in diverse eukaryotes [1–5]. Amplification of both driving oncogenes and chemotherapy

**Funding:** Funding for JH, RH, MK and GP was from the Wellcome Trust [088335,110216], wellcome.ac.uk, for JH and FK from the Biotechnology and Biological Sciences Research Council BBSRC [BI Epigenetics ISP: BBS/E/B/000C0423], https://bbsrc.ukri.org/, and for XV from the European Commision [Erasmus+ programme], www.erasmusplus.org.uk. The funders had no role in study design, data collection and analysis, decision to publish, or preparation of the manuscript.

**Competing interests:** The authors have declared that no competing interests exist.

**Abbreviations:** CEN, centromere; DSB, double strand break; eccDNA, extrachromosomal circular DNA; ERC, extrachromosomal ribosomal DNA circle; ExoV, exonuclease V; HR, homologous recombination; LINE, Long interspersed nuclear element; MEP, mother enrichment program; NHEJ, Non-homologous end joining; rDNA, ribosomal DNA; REC-seq, restriction-digested extrachromosomal circular DNA sequencing; SSA, single strand annealing; ssDNA, single-stranded DNA.

resistance genes on eccDNA has also been widely reported in tumour cells and is a frequent feature of cancer genomes [6,7]. Genome-wide studies in yeast, worms, and nontransformed mammalian cells reveal huge diversity in the protein coding eccDNA complement [8–10], in addition to widespread microDNA, heterochromatic repeat-derived eccDNA, and telomeric circles [11–13]. Nevertheless, we understand little of the mechanisms of protein-coding eccDNA formation because such events are generally rare and unpredictable.

Reintegration of eccDNA provides an efficient pathway for gene amplification, with various chromosomal adaptations in lower eukaryotes being known or predicted to have emerged through eccDNA intermediates [14–18]. In tumour cells, aggregation of smaller eccDNA to large (1–5 MB), microscopically visible double minutes and reintegration in homogeneously staining regions to yield high-copy chromosomal repeats has also been reported [7,19,20]. Thereby, eccDNA acts as an intermediate in chromosome structural evolution. However, eccDNA usually segregates randomly during mitosis because of the absence of a centromere, causing pronounced heterogeneity in eccDNA complement across the population that provides dramatic phenotypic plasticity [21]. Theoretical models predict that gene copy number amplification can occur much faster via eccDNA accumulation than on chromosomal DNA [7], and population heterogeneity in eccDNA gene dosage can be selected not only for increased gene dosage [21,22] but also for decreased dosage because of the ease of eccDNA loss in cell division [18,23,24].

Segregation of eccDNA during cell division is not necessarily random [25]. This is exemplified by budding yeast replicative ageing wherein cells divide asymmetrically into mother and daughter cells, with mothers retaining various molecules that could be detrimental to daughter cell fitness, including a well-studied eccDNA species—extrachromosomal ribosomal DNA circles (ERCs) [26,27]. Retention of ERCs in mother cells involves tethering by SAGA and TREX2 to nuclear pore complexes, which are themselves largely retained in mother cells, with mutation of SAGA components abrogating retention and extending mother cell life span [28–30]. ERCs replicate on each division, and mother cell retention leads to exponential copy number amplification, increasing genome size by 50% in 24 hours [31], which is thought to interfere with various critical pathways to accelerate, if not cause, ageing pathology [32,33].

Formation mechanisms for eccDNA have been primarily inferred from amplicon structure. The conceptually simple episome model invokes a recombination event between distal sites on the same chromosome to yield an eccDNA with a matching deletion [19]. Chromosomal deletions closely matching eccDNA breakpoints have been observed, although not universally [3,19,34–37]. However, recombination would be expected to favour sites with substantial homologous sequence, and although this is observed at breakpoints in some systems, microhomology is commonly reported [3,9,12,22,37,38]. Conversely, replication-based models allow eccDNA formation without chromosomal change. Substantial evidence supports the formation of ERCs in yeast from replication forks stalled at the ribosomal DNA (rDNA) replication fork barrier [39,40]; however, it is unclear whether such events are specific to the yeast rDNA, which contains a unique replication fork stalling system [41,42]. Other mechanisms invoke re-replication of DNA to yield nested structures that must be resolved by homologous recombination (HR) prior to mitosis [43] and can potentially yield eccDNA of great complexity. Such models predict no clear structural signature and have therefore been hard to test, although de-repression of origin relicensing does lead to chromosomal copy number variation as predicted [44]. Without reproducible methods to observe eccDNA formation de novo, it remains very difficult to validate these models or probe mechanistic details.

A few examples of regulated eccDNA formation have been reported. Stage specific de novo formation of eccDNA during embryonic development in *Xenopus* facilitates rolling circle amplification of the rDNA [45,46], and eccDNA production in T- and B cells as a side-effect of

highly-regulated V(D)J recombination is well-studied [47]. In yeast, rDNA recombination including ERC formation is highly dependent on histone deacetylases that respond to nutrient availability [4,48,49], leading us to question whether protein-coding eccDNA formation could also be driven by environmentally responsive signalling, analogous to the transcriptionally stimulated chromosomal copy number variation that we and others recently reported [50,51]. Here, we describe the transcriptionally stimulated formation of eccDNA from the yeast *CUP1* locus and characterise the mechanism by which eccDNA is processed from DSBs.

## Results

### Transcription promotes eccDNA accumulation from the CUP1 locus

ERCs are enriched in aged yeast, and we speculated that other eccDNA may be similarly enriched during ageing. To obtain highly purified aged yeast samples, we employed the mother enrichment program (MEP) combined with cell wall labelling [52,53]. Young cells in log phase growth are labelled with a cell wall reactive biotin derivative that is completely retained on mother cells because daughter cell walls are newly synthesised on each division [53]. Estradiol is added to activate the MEP, after which all newborn daughter cells become nonreplicative. Mother cells are therefore the only replicative cells in the population and reach advanced age without the culture becoming saturated; these highly aged mother cells can then be affinity purified (Fig 1A).

To test the potential for environmentally stimulated eccDNA accumulation, we analysed the locus encoding the copper-resistance gene *CUP1*. Exposure to environmental copper strongly induces transcription of the *CUP1* gene, which is encoded in an array of 2 to 15 2 kb tandem repeats on Chromosome VIII [54,55]. A labelled cohort of cells was aged in the presence or absence of 1 mM $CuSO_4$ for 48 hours (Fig 1B lanes 3,4), and control cells were maintained at log phase for the same period (Fig 1B lanes 1,2). Southern blot analysis of aged cohorts revealed multiple DNA species in the copper-treated sample migrating above the resolution limit of the gel, a region that is largely devoid of linear DNA (Fig 1B lane 3). These were much less prominent for cells aged in the absence of copper and undetectable in log phase cells (Fig 1B lanes 1–4).

To confirm that these signals represent eccDNA, genomic DNA from cells aged in 1 mM $CuSO_4$ was digested with exonuclease V (ExoV), which degrades linear DNA progressively from double stranded ends but has no activity on circular or nicked circular DNA. As expected, ExoV efficiently degraded DNA from the chromosomal *CUP1* band, whereas the species above the resolution limit were unaffected (S1A Fig). Copper-stimulated formation of *CUP1* eccDNA was reproducible and significant and did not represent a general change in eccDNA accumulation because ERCs were detected at a slightly, but not significantly, higher level in cells aged in the absence of copper (Fig 1B, lanes 5–8). A time course analysis was then performed using genomic DNA treated with ExoV, which increases sensitivity by removing chromosomal DNA signals, revealing that *CUP1* eccDNA is readily detected after 24 hours in copper, but not after 7.5 hours, so eccDNA accumulation increases with age (Fig 1C). Although eccDNA signals are stronger after 48 hours, we have previously noted that $Cu^{2+}$ reduces cell viability during ageing [50], and we therefore restricted further experiments involving ageing cells in $CuSO_4$ to 24 hours.

Accumulation of *CUP1* eccDNA may offer a selective advantage by increasing copper resistance. To ensure that selection for copper resistance does not underlie the observed eccDNA accumulation, we performed a similar experiment using a strain in which transcription at the *cup1* locus is induced under a different environment and does not yield active protein. We have previously developed a yeast strain in which all *CUP1* repeats have been mutated such

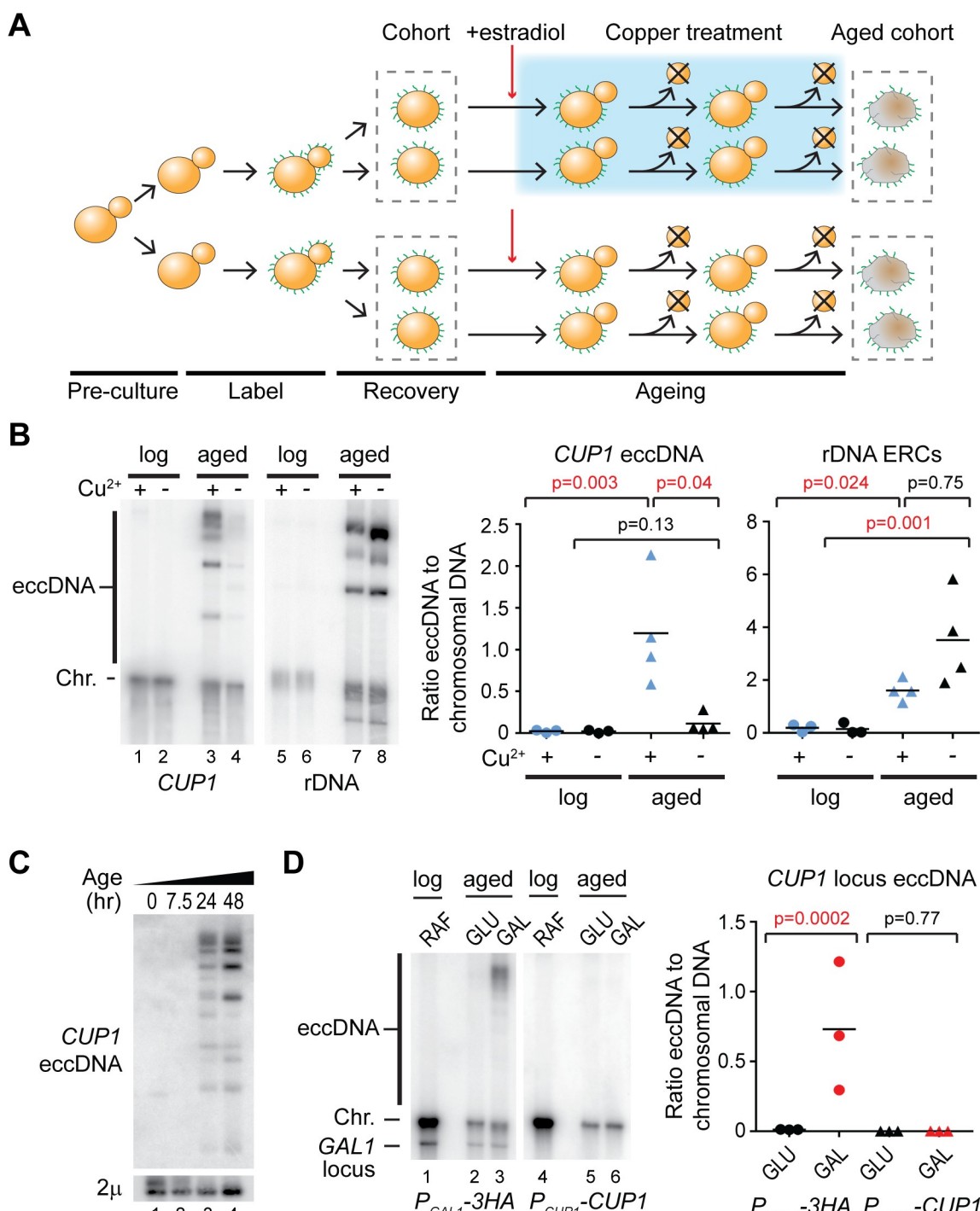

**Fig 1. Transcription of the *CUP1* locus causes eccDNA accumulation.** (A) Schematic representation of cell labelling, induction, and ageing in the presence or absence of copper. (B) Southern blot analysis of *CUP1* eccDNA and rDNA-derived eccDNA (ERCs) in yeast cells aged for 48 hours in the presence or absence of 1 mM $CuSO_4$, along with young cells maintained in log phase in the presence or absence of 1 mM $CuSO_4$ for an equivalent time. Large linear fragments of chromosomal DNA (Chr.) migrate at the resolution limit of the gel, whereas circular DNA species (eccDNA) migrate more slowly. Abundances of eccDNA and ERCs were compared by one-way ANOVA; *n* = 4 biological replicates; data were log transformed for testing to fulfil the assumptions of a parametric test. (C) Southern blot analysis of *CUP1* eccDNA in cells aged for 0, 7.5, 24, or 48 hours in 1 mM $CuSO_4$. Chromosomal DNA was removed with ExoV to improve sensitivity, endogenous circular 2µ DNA is shown as a loading control. (D) Southern blot analysis of eccDNA in a heterozygous strain bearing one $P_{GAL1}$-*3HA cup1* locus modified by replacing all *CUP1* promoters and ORFs with $P_{GAL1}$ promoters and *3HA* ORFs, and 1 wild-type $P_{CUP1}$-*CUP1* allele. eccDNA is detected with allele specific probes; the additional band in the left panel

is from hybridisation to the endogenous *GAL1* locus on Chromosome II. Quantification and analysis performed as in panel B, *n* = 3. The data underlying this figure may be found in S1 Data and S1 Raw Images. eccDNA, extrachromosomal circular DNA; ERC, extrachromosomal ribosomal DNA circle; ExoV, exonuclease V; ORF, open reading frame; rDNA, ribosomal DNA.

that the *CUP1* protein-coding sequences are replaced with a nonfunctional *3HA* sequence and the $P_{CUP1}$ promoters are replaced by galactose-responsive *GAL1* promoters ($P_{GAL1}$) but other sequences in the *CUP1* repeat are retained (S1B Fig) [50]. These cells transcribe mRNA encoding only a nonfunctional 3HA protein from every *cup1* repeat in the presence of galactose but not glucose. Here, we mated this strain with a wild-type haploid to form a heterozygous diploid with one galactose-responsive $P_{GAL1}$-*3HA cup1* allele and one copper-responsive $P_{CUP1}$-*CUP1* allele. When aged in galactose, we observed that these cells form eccDNA from the $P_{GAL1}$-*3HA* allele but not from the copper-responsive wild-type $P_{CUP1}$-*CUP1* allele, whereas no eccDNA from either allele was detectable in cells aged in glucose (Fig 1D, compare lanes 2, 3 and 5, 6).

These experiments show that transcriptional activiation of the *CUP1* locus causes the accumulation of eccDNA in aged cells even in the absence of selection and that eccDNA accumulation is specific to the transcriptionally induced locus versus a noninduced control locus.

## eccDNA accumulation is locus specific

To reveal eccDNA distribution across the genome, we developed a quantitative eccDNA sequencing protocol for the approximately 30 ng genomic DNA available from our standard ageing yeast preparations. Previous eccDNA sequencing analysis in yeast has involved exonuclease digestion and rolling circle amplification [9]. The latter reaction is affected by circle size, making quantitative comparisons between circular species challenging. Alternative biochemical methods are quantitative but require a prohibitive amount of starting material [10]. We therefore employed extensive exonuclease treatment of aged genomic DNA followed by DNA fragmentation and a sensitive library preparation protocol to reveal the circular DNA complement (S2A Fig).

Exonuclease treatment alone enriched for ERCs and subtelomeric eccDNA (S2A Fig); however, >95% of sequencing reads in these libraries mapped to rDNA and 2μ, which limited sensitivity. To remove these species, each sample was split in 3 parts and treated with different restriction enzymes prior to exonuclease treatment. All 3 restriction enzymes cleave ERCs and 2μ, rendering both sensitive to exonuclease digestion, whereas any circle that does not contain all 3 restriction sites will be maintained in at least one reaction (S2B Fig). After exonuclease treatment, the 3 parts were reunited for sequencing. This restriction-digested extrachromosomal circular DNA sequencing protocol (REC-seq; Fig 2A) reduced reads from ERCs and 2μ to <2%, revealing eccDNA from other regions, including Ty elements, *CUP1*, the *ENA* locus and *HXT6-HXT7*, all of which have been previously described using less quantitative methods (S2B Fig) [9].

In agreement with Southern blotting data, REC-seq on cells aged in the presence of copper showed a substantial (approximately 10-fold) increase in reads mapping to the *CUP1* locus but not to other loci, such as the subtelomeric regions (Fig 2B). The *CUP1* eccDNA reads cover precisely the *CUP1* repeated region, strongly suggesting formation through an HR mechanism. Read quantification across the genome in 20 bp windows revealed that although eccDNA accumulates from many loci in aged cells, only the *CUP1* locus is differentially affected after 24 hours (Fig 2C). Applying the same method to the heterozygous $P_{GAL1}$-*3HA*/$P_{CUP1}$-*CUP1* strain aged for 48 hours in galactose or glucose media revealed many more $P_{GAL1}$-*3HA cup1* locus reads in cells aged on galactose, with a pronounced dip in the profile at the region of the repeat

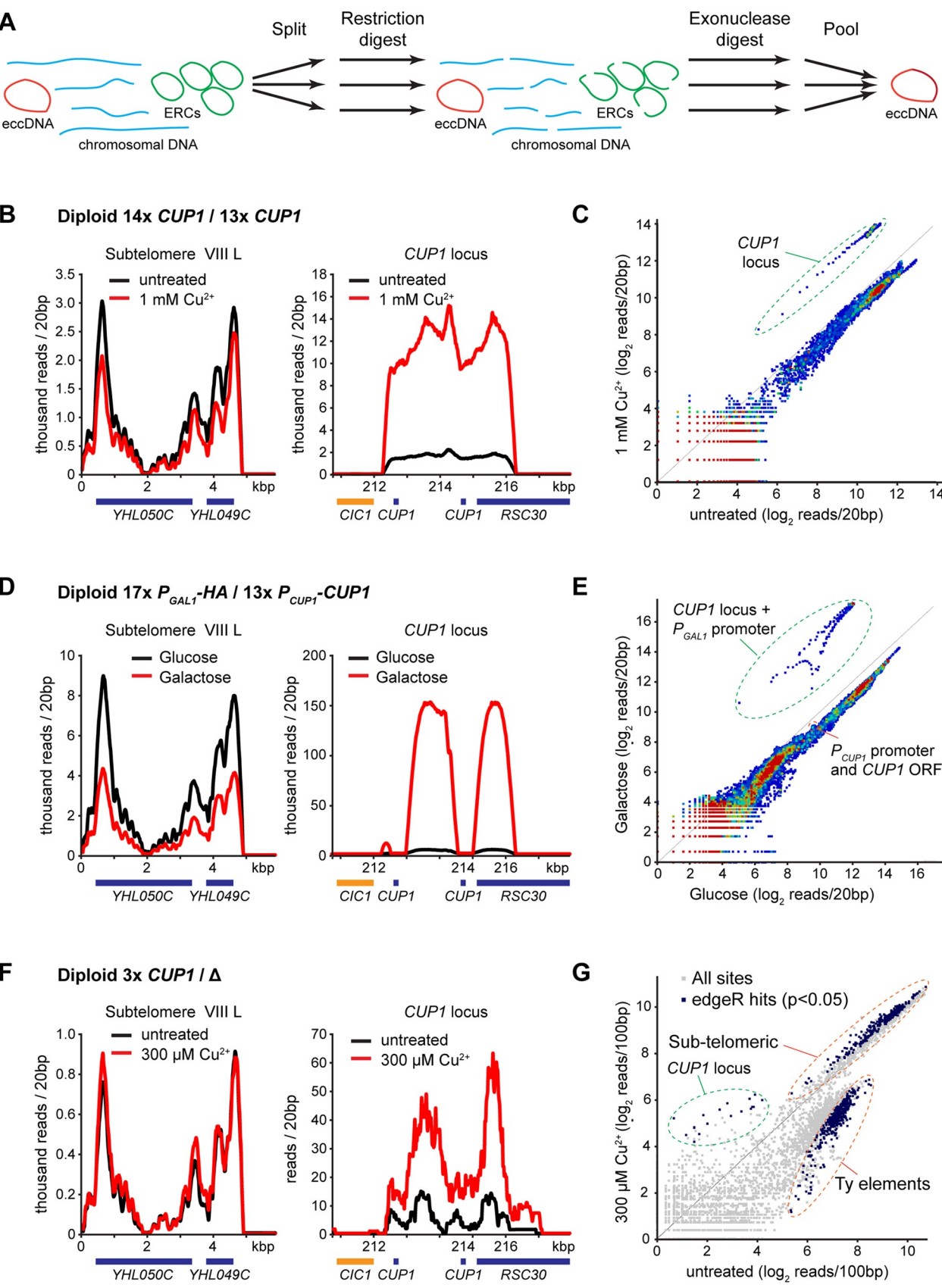

**Fig 2. Genome-wide quantitative mapping of eccDNA in aged cells.** (A) Schematic representation of library preparation method. Briefly, genomic DNA is split into 3 samples and digested with 3 different endonucleases that all target ERCs and 2μ DNA and then is digested with ExoV and ExoI. After 2 rounds of treatment, samples are reunited, fragmented, and a DNA sequencing library prepared by standard methods. (B) Distribution of sequencing reads around the *CUP1* locus on Chromosome VIII and Subtelomere VIII L obtained from REC-seq of cells aged for 24 hours in the presence or absence of 1 mM $CuSO_4$. (C) Scatter plot of read counts in 20 bp bins across the genome for samples shown in panel B. Circled points represent bins within the *CUP1* repeat region. (D and E) Equivalent analysis to panels B and C performed on $P_{GAL1}$-*3HA*/$P_{CUP1}$-*CUP1* heterozygous strain aged for 48 hours in glucose or galactose. The pronounced dip in reads mapping to the *CUP1* locus over the actual *CUP1* ORFs corresponds to the region of the *CUP1* repeat replaced with $P_{GAL1}$-*3HA* and demonstrates that essentially all the eccDNA is formed from the $P_{GAL}$-*3HA* allele. Bins representing the *CUP1* promoter and ORF are also highlighted in panel E. (F) Analysis of reads mapping to *CUP1* locus and Subtelomere VIII L averaged across 3 biological replicates of 3x*CUP1*/Δ cells aged for 24 hours in the presence or absence of 300 μM $CuSO_4$. (G) Scatter plot of average read counts across 3 biological replicates in 100 bp bins across the genome for samples shown in panel F. Blue dots are significantly different between cells aged in the presence or absence of copper based on edgeR analysis; $p < 0.05$. The genomic location of the different clusters of significantly different bins are indicated; all those within the *CUP1* locus area are derived from *CUP1* and are the most enriched in the presence of copper. The data underlying this figure may be found in S1 Data. eccDNA, extrachromosomal circular DNA; ERC, extrachromosomal ribosomal DNA circle; ExoV, exonuclease V; ORF, open reading frame; REC-seq, restriction-digested extrachromosomal circular DNA sequencing.

containing the *CUP1* promoter and ORF, both of which are absent from the $P_{GAL1}$-*3HA* allele but present in the $P_{CUP1}$-*CUP1* allele (S1B Fig, Fig 2D). Again, by scatter plot, the specificity of eccDNA accumulation from the mutated $P_{GAL1}$-*3HA cup1* allele was obvious, whereas reads mapping to the *CUP1* promoter and ORF that are only present on the wild-type $P_{CUP1}$-*CUP1* allele were not differentially enriched (Fig 2E). Therefore, eccDNA accumulation from the *CUP1* locus promoted by transcription is remarkably locus specific.

REC-seq is more sensitive than Southern blot, allowing quantification of *CUP1* eccDNA accumulation in a strain with only 3 tandem copies of the *CUP1* locus; this represents a more general case because loci containing small numbers of tandem repeats are common in eukaryotic genomes [56,57]. eccDNA from *CUP1* accumulated more in cells aged in copper (Fig 2F); the difference (approximately 4-fold) was smaller than in the MEP wild-type strain, but this is not unexpected given that the 3x*CUP1*/Δ cells contain 9-fold fewer copies of *CUP1*. The genome-wide profile of eccDNA was more affected by copper treatment in these cells, likely due to greater copper stress, and many eccDNAs accumulated to lower levels (Fig 2G). However, edgeR analysis of differential accumulation across 3 biological replicates revealed that the significant regions most over-represented in the copper-treated sample all derive from the *CUP1* locus (Fig 2G blue dots; $p < 0.05$ by EdgeR). Additionally, eccDNA from many Ty-element loci was significantly under-accumulated in copper-treated cells, whereas eccDNA from some subtelomeric sites was significantly overaccumulated though by a very small amount (on average $0.3 \log_2$ units).

Therefore, eccDNA accumulation caused by transcription is remarkably site specific and is observable even in low-copy tandem repeats.

## Asymmetric segregation is necessary but not sufficient for eccDNA accumulation

Asymmetric segregation of replicating ERCs causes ERC accumulation during ageing. The retention of ERCs in mother cells depends on attachment to the nuclear pore complex through the SAGA chromatin modifying complex, and rapid progression from S phase through mitosis so that ERCs are not passed to daughter cells by diffusion [28,29]. We found that cells lacking the SAGA component Spt3 contain far less *CUP1* eccDNA and ERCs after 24 hours (Fig 3A and 3B), even though transcriptional induction of the *CUP1* promoter was normal in *spt3*Δ cells (S3 Fig). We observed a similar defect in *bud6*Δ and *yku70*Δ mutants that are defective in ERC asymmetric segregation due to bud neck formation issues or extended G2, respectively [29,58] (Fig 3A). Therefore, asymmetric segregation is necessary for *CUP1* eccDNA accumulation.

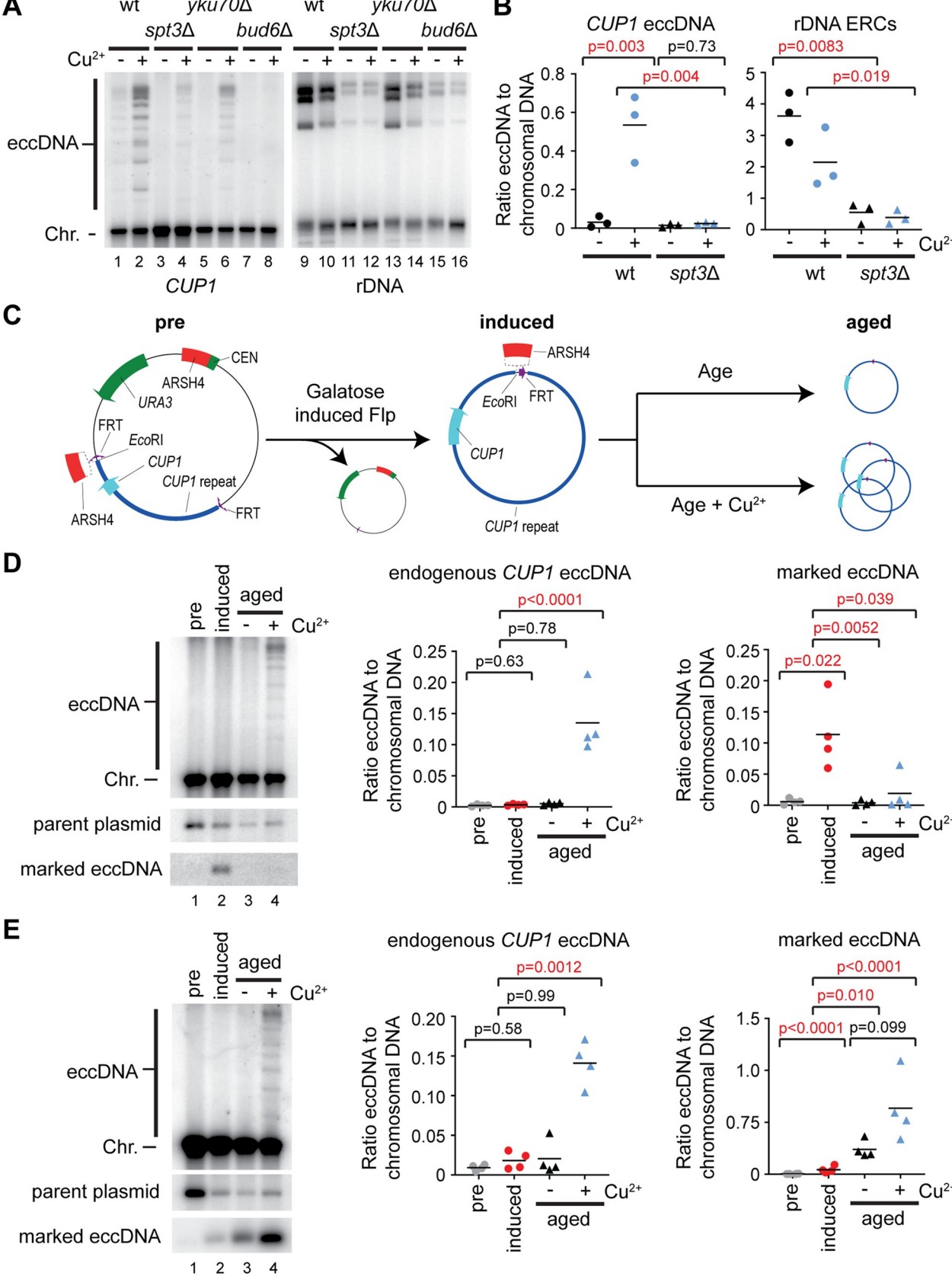

**Fig 3. Asymmetric inheritance is necessary but not sufficient for eccDNA accumulation.** (A) Southern blot showing *CUP1* eccDNA and ERCs for wt, *spt3Δ*, *yku70Δ*, and *bud6Δ* cells aged for 24 hours in the presence or absence of 1 mM CuSO₄. (B) Quantification of *CUP1* eccDNA and rDNA-derived ERCs in wt and *spt3Δ* cells aged for 24 hours in the presence or absence of 1 mM CuSO₄, performed and analysed as in Fig 1B, n = 3. (C) Schematic representation of method to form and quantify marked *CUP1* eccDNA. Briefly, galactose-inducible Flp recombinase is used to excise a marked *CUP1* eccDNA with an *Eco*RI site absent from endogenous *CUP1* repeat region. Distribution of the marked eccDNA and endogenous species after 24 hours of ageing can be quantified by Southern blot. A variant carrying an additional ARSH4 sequence within the marked eccDNA was also constructed as shown. (D) Marked eccDNA analysis: MEP wt cells cured of 2μ were transformed with one plasmid carrying a *CUP1* repeat and an *Eco*RI site flanked by FRT sites and a second plasmid expressing Flp from a galactose-responsive promoter. Cells were grown overnight in selective media to maintain plasmids and containing sucrose and raffinose as carbon sources (lane 1, pre). Flp expression was induced by addition of galactose for 4 hours, then cells were labelled, inoculated in 3 cultures, and grown in nonselective glucose media for 2 hours after which 1 culture was harvested (lane 2, induced); 1 mM CuSO₄ was then added to 1 culture, and both cultures were grown for 24 hours at 30°C before harvesting (lanes 3–4, aged). *Eco*RI-digested genomic DNA was analysed by Southern blot and probed for the *CUP1* repeat to reveal all species shown. Marked and endogenous *CUP1* eccDNA was quantified; band intensities were log transformed to fulfil the assumptions of a parametric test and analysed by one-way ANOVA, n = 4 biological replicates. (E) Modification of marked eccDNA experiment: the ARSH4 origin of replication was cloned into the marked *CUP1* plasmid between the *Eco*RI site and the *CUP1* repeat sequence such that ARSH4 remains in the marked *CUP1* eccDNA after Flp recombination (see schematic in panel C). Experiment was performed exactly as in panel D. The data underlying this figure may be found in S1 Data and S1 Raw Images. eccDNA, extrachromosomal circular DNA; ERC, extrachromosomal ribosomal DNA circle; FRT, flippase recognition target; MEP, mother enrichment program; rDNA, ribosomal DNA; wt, wild type.

We considered 2 mechanisms by which asymmetric segregation could be important for eccDNA accumulation. Firstly, the small population of *CUP1* eccDNA known to exist in young cells could be maintained by asymmetric segregation and amplified by replication because the *CUP1* repeat contains a replication origin [9]; this model, which is equivalent to the ERC amplification pathway, requires efficient replication that exceeds the loss rate of eccDNA by segregation to daughter cells. Alternatively, eccDNA could form frequently and be further concentrated in aged cells by asymmetric segregation; this model does not require efficient replication of eccDNA as long as the de novo formation rate exceeds the loss rate.

To directly test whether *CUP1* eccDNA can be maintained and amplified across ageing, we designed an inducible system to generate marked *CUP1* eccDNA in young cells (Fig 3C). A single *CUP1* repeat with an additional *Eco*RI site absent from the endogenous *CUP1* sequence was flanked by FRT sites for Flp recombinase and cloned into a stable centromeric plasmid. This was transformed into a MEP strain previously cured of 2μ (a parasitic plasmid that constitutively expresses Flp), along with a second plasmid expressing Flp under a galactose-responsive promoter. Cells were pregrown in raffinose/sucrose (Fig 3C, pre) before a 4 hour pulse with galactose to induce Flp expression and form the marked *CUP1* eccDNA (Fig 3C, induced). Labelling and recovery in glucose were then performed as normal before estradiol addition and ageing for 24 hours in glucose media with or without copper (Fig 3C, aged). After the recovery period, 11% ± 6% of the plasmid had recombined to form the marked *CUP1* eccDNA, a low but unambiguously detectable level (Fig 3D, lane 2). After 24 hours of ageing in the presence or absence of copper, the marked eccDNA was undetectable in almost all samples, despite a clearly detectable accumulation of eccDNA from the endogenous chromosomal *CUP1* locus (Fig 3D, lanes 3 and 4). This shows that *CUP1* eccDNA is not perfectly retained during ageing, and so in the absence of de novo formation, *CUP1* eccDNA is lost.

If asymmetric segregation is sufficient for ERC accumulation, why is *CUP1* eccDNA different? One possibility is that the origins of replication in each *CUP1* repeat are inefficient. We therefore constructed a variant of the marked *CUP1* eccDNA containing a high-efficiency ARSH4 (ARS209) origin (Fig 3C). In contrast to native *CUP1* eccDNA, this ARS-*CUP1* eccDNA was maintained and amplified across ageing (Fig 3E), showing that the defect in *CUP1* eccDNA maintenance stems from the replication efficiency of the eccDNA being lower than the rate of loss to daughter cells. The ARS-*CUP1* eccDNA was consistently amplified to higher levels in cells aged in the presence of copper, although this effect did not reach significance; this is coherent with the requirement for SAGA component Spt3 in *CUP1* eccDNA

maintenance (Fig 3A and 3B), because Spt3 is known to be recruited to the *CUP1* promoter on transcriptional induction, as well as to the *GAL1* promoter [59,60]. However, this difference is small (approximately 2-fold) relative to the differential accumulation of endogenous *CUP1* eccDNA in the presence or absence of copper (approximately 10-fold).

Together these experiments show that *CUP1* eccDNA is subject to asymmetric segregation, but this is insufficient for maintenance because *CUP1* eccDNA is not efficiently replicated. *CUP1* eccDNA must therefore be formed at a high enough rate during ageing to offset losses through segregation to daughter cells.

## CUP1 eccDNA is formed through DSB processing by Sae2 and Mre11

Various mechanisms have been proposed for eccDNA formation involving repair of single strand breaks or DSBs that are either linked to DNA replication or not (reviewed in the work by Paulsen and colleagues [61]). To delimit the eccDNA formation mechanism, we analysed eccDNA accumulation in a selection of DSB processing mutants.

Many DSB processing mutants have shortened lifespans and cannot be aged sufficiently to show eccDNA accumulation [62,63]; however, we characterised a small panel of informative DSB repair mutants (*dnl4Δ*, *exo1Δ*, *sae2Δ*) that aged effectively in the MEP system (Fig 4A). Firstly, loss of Dnl4, a critical DNA ligase for nonhomologous end joining (NHEJ), did not substantially alter *CUP1* eccDNA or ERC levels (compare lanes 4 and 12 to wild-type controls 2 and 10). Secondly, loss of exonuclease Exo1, which is involved in long-range DNA end resection as well as degradation of stalled or reversed replication forks, had no effect on *CUP1* eccDNA but substantially increased ERC levels (compare lanes 6 and 14 to wild-type controls). Thirdly, the absence of end-resection factor Sae2 caused a major reduction in *CUP1* eccDNA but had little effect on ERCs (compare lanes 8 and 16 to wild-type controls).

Sae2 is important for initiating DNA end resection from DSBs, particularly when DNA ends are blocked or cannot be processed (reviewed in the work by Symington [64]), and the importance of this factor for *CUP1* eccDNA formation suggests a formation mechanism involving DNA damage. In contrast, ERCs are known to form through a replication fork stalling mechanism that is consistent with the increased ERC levels in *exo1Δ* cells in which stalled fork degradation is impaired. Therefore, the observation that *CUP1* eccDNA but not ERC accumulation is dependent on Sae2 strongly suggests that these species do not form by the same mechanism in aged cells. Usefully, the normal accumulation of ERCs in *sae2Δ* demonstrates that reduced *CUP1* eccDNA cannot simply be attributed to an ageing defect, because this would also diminish ERC levels. To ensure that this critical phenotype was reproducible, we generated 2 further *sae2Δ* mutants in the MEP background with different markers and consistently observed the same phenotype (Fig 4B).

Sae2 mediates end resection, stimulating the nuclease activity of Mre11 to generate a short single-stranded 3' end and allows access to long-range resection activities of Exo1 and Dna2/Sgs1. Mre11 is a member of the Mre11-Rad50-Xrs2 (MRX) complex, and life span is severely compromised in *mre11Δ* cells just as for previously described *rad50Δ* mutants [63]. However, MRX has important functions in DSB repair beyond resection, and we observed that cells carrying the nuclease deficient *mre11*[H125N] allele behaved similarly to wild type during ageing. Importantly, *mre11*[H125N] cells are defective in *CUP1* eccDNA accumulation but have little defect in ERC accumulation (Fig 4C), mirroring the *sae2Δ* phenotype (Fig 4A and 4B). *sgs1Δ* and particularly *sgs1Δ exo1Δ* mutants show variable but severe life span defects and therefore the importance of long-range resection could not be assessed. Nonetheless, these results show that DSB formation and resection are critical steps in *CUP1* eccDNA formation.

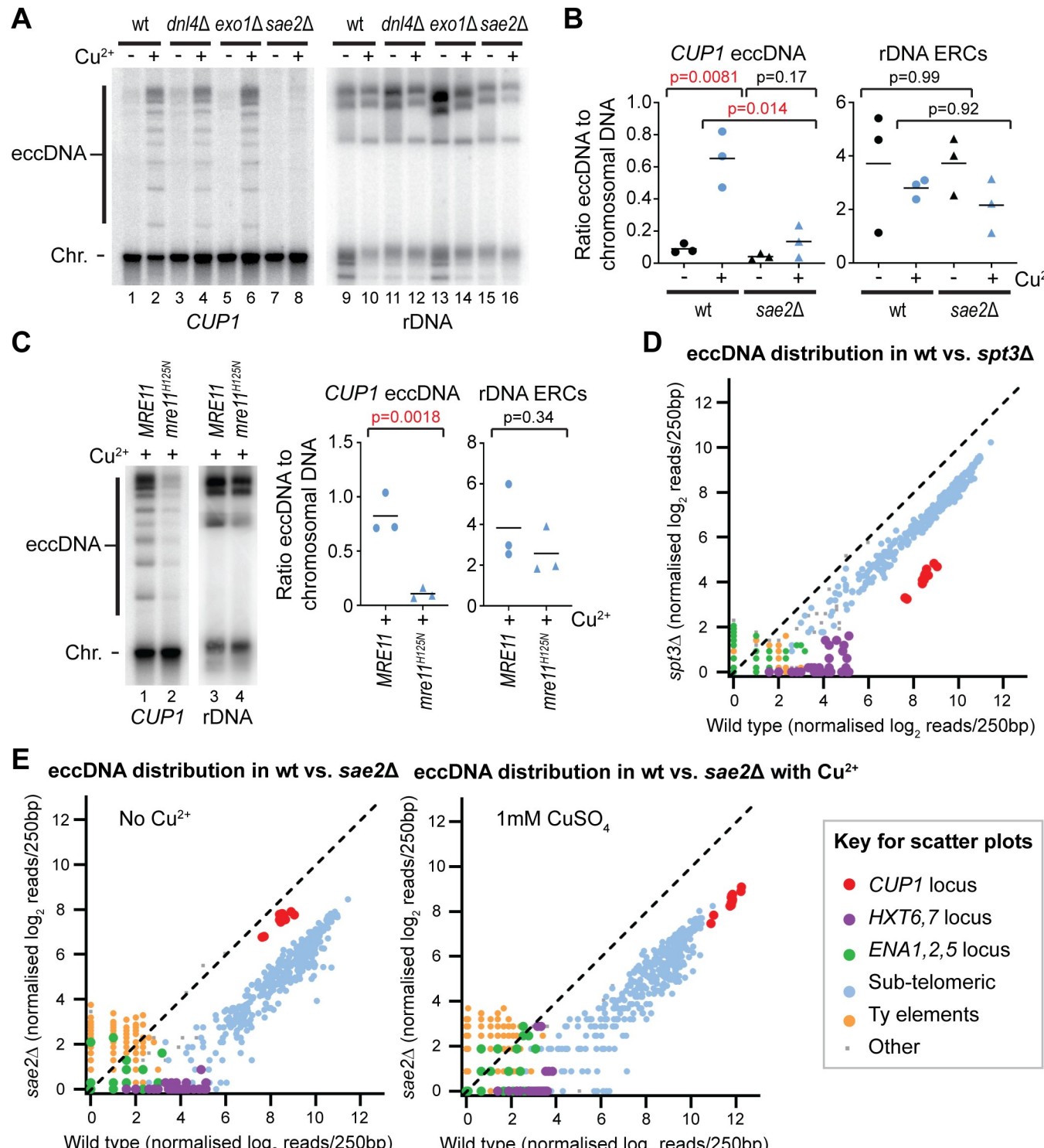

**Fig 4. Sae2 and Mre11 nuclease activity are required for eccDNA formation.** (A) Southern blot analysis of *CUP1* eccDNA and rDNA-derived ERCs in wt, *dnl4Δ*, *exo1Δ*, and *sae2Δ* cells aged for 24 hours in the presence or absence of 1 mM CuSO₄, performed as in Fig 1B. (B) Analysis of *CUP1* eccDNA and rDNA-derived ERCs in wt and *sae2Δ* cells aged for 24 hours in the presence or absence of 1 mM CuSO₄, performed and analysed as in Fig 1B; *n* = 3; quantification is derived from 3 separate clones of *sae2Δ* with 3 different selectable markers. (C) Analysis of *CUP1* eccDNA and rDNA-derived ERCs in wt and *mre11*^H125N^/ *mre11Δ* aged for 24 hours in the presence of 1 mM CuSO₄, performed and analysed as in Fig 1B, *n* = 3. (D) REC-seq analysis comparing wt and *spt3Δ* cells aged for 24 hours. Samples were digested with restriction enzymes that leave the 2μ DNA intact, and then read counts were normalised based on 2μ counts in eccDNA and matching total DNA libraries such that eccDNA counts are relative to an equivalent quantity of total chromosomal DNA in each sample. Reads are

quantified in 100 bp windows spanning the genome (excluding rDNA, mitochondria, 2μ, and *UBC9* locus), and coloured by feature according the key below. (E) REC-seq analysis of *sae2Δ* cells compared to wt in the absence (left) or presence (right) of 1 mM CuSO$_4$. Experiment and analysis as in panel D. The data underlying this figure may be found in S1 Data and S1 Raw Images. eccDNA, extrachromosomal circular DNA; ERC, extrachromosomal ribosomal DNA circle; rDNA, ribosomal DNA; REC-seq, restriction-digested extrachromosomal circular DNA sequencing; wt, wild type.

To reveal the locus specificity of Sae2 function in eccDNA formation, we performed REC-seq in *sae2Δ* mutants and also in *spt3Δ* mutants to allow comparison between defects in eccDNA formation and retention. Global effects on eccDNA levels are not captured by our original REC-seq method because there is no way to normalise read counts between samples. To provide a fixed point for normalisation, we replaced one of the restriction enzymes in the REC-seq protocol with *Sma*I, which cleaves the rDNA but not the 2μ element; 2μ is a high-copy circular plasmid present in all laboratory yeast strains and can be readily detected in both eccDNA and total input DNA sequencing data. We sequenced total DNA and eccDNA from each sample and normalised eccDNA to nonrepetitive chromosomal DNA using 2μ read counts as a fixed point in a manner that preserves differences in 2μ levels between samples (the strategy is depicted in S4 Fig). This modification entails a compromise because 2μ reads make up approximately 90% of the read count in these REC-seq libraries, dramatically reducing the signal to noise ratio from other regions; however, most eccDNA species remain detectable.

Using this modified REC-seq method, we observe that all eccDNA species are under-represented in aged *spt3Δ* cells, albeit to a variable extent, with eccDNA from the *CUP1* and *HXT6-HXT7* loci showing the greatest difference (>10 fold), whereas subtelomeric circles and Ty elements are more modestly affected (approximately 4-fold; Fig 4D). SAGA has been previously shown to bind promoters including *CUP1* on transcriptional activation [59,65]; however, the importance of SAGA for mother cell retention of *CUP1* eccDNA in the absence of copper, as well as in retaining diverse other eccDNA species, suggests that SAGA acts on eccDNA irrespective of transcription.

In contrast, the eccDNA accumulation phenotype of *sae2Δ* cells is complex. Wild-type cells aged in the presence of copper contain approximately 8-fold more *CUP1* eccDNA than *sae2Δ* cells (Fig 4E right panel, red spots), which is in accord with the Southern blot data (Fig 4A and 4B). However, this enrichment is not observed for cells aged in the absence of copper (Fig 4E left panel, red spots). This interesting difference suggests that the Sae2-dependent mechanism of *CUP1* eccDNA formation is initiated by transcription, but that a background Sae2-independent mechanism forms *CUP1* eccDNA at a basal level in uninduced cells. We also note that Ty-element eccDNA is not reduced in the absence of Sae2, consistent with the previously proposed mechanism of Ty-element eccDNA formation through direct recombination between terminal repeat regions [66] (Fig 4E, orange dots). However, eccDNA from subtelomeric circles, the *HXT6-HXT7* locus, and to some extent the *ENA1-ENA2-ENA5* locus, are also Sae2 dependent, suggesting that this mechanism of eccDNA formation acts at multiple loci (Fig 4E blue, purple, and green dots, respectively).

These data reveal that multiple eccDNA species form through a DSB processing pathway involving Sae2, including the transcriptionally induced *CUP1* eccDNA. However, a Sae2-independent pathway is also active, giving rise to ERCs as well as the basal level of *CUP1* eccDNA in non–copper-treated cells.

## Mus81 is required for CUP1 eccDNA formation in young and aged cells

DSB resection alone does not yield eccDNA, which most likely requires intrachromosomal strand invasion at a homologous sequence (notably all the eccDNA species reproducibly detected by REC-seq are bounded by homologous regions). This recombination intermediate

must be actively processed by resolvase enzymes to release circular DNA. We therefore asked which of the 3 known yeast resolvase enzymes Mus81-Mms4, Yen1, or Slx1-Slx4 are required. *mus81Δ* cells aged in copper had dramatically reduced *CUP1* eccDNA compared to wild type, whereas *CUP1* eccDNA levels in *yen1Δ* and *slx4Δ* mutants were indistinguishable from wild type, indicating that Mus81 is the primary resolvase for eccDNA formation (Fig 5A, compare lanes 2,4).

Reduced eccDNA formation in *mus81Δ* was highly reproducible (Fig 5B); however, we also observed a pronounced reduction in ERC levels (Fig 5A lanes 9–12 and Fig 5B), and bud-scar counting revealed that *mus81Δ* cells aged in copper are younger than wild-type cells (wild-type + $Cu^{2+}$: 12.2 ± 3.7 versus *mus81Δ* + $Cu^{2+}$: 10.5 ± 4.2), raising questions as to whether the observed *mus81Δ* phenotype actually stems from an ageing defect. Interestingly, REC-seq data comparing *mus81Δ* cells to wild type was very similar to that of *sae2Δ* cells (compare Fig 5C to Fig 4E), showing a reduced accumulation of *CUP1*, subtelomeric circles and other species except for Ty-element eccDNA, consistent with both proteins acting in the same pathway of eccDNA formation. The only notable difference is that *CUP1* eccDNA in the absence of copper is less abundant in *mus81Δ* than in wild-type cells, whereas the same is not true for *sae2Δ*.

The similarity between *mus81Δ* and *sae2Δ* REC-seq profiles led us to question the contribution of ageing to eccDNA accumulation phenotypes given that *mus81Δ* has a slow ageing phenotype. In theory, lower eccDNA levels may result not only from reduced formation, but also from cells being younger when sampled or, more subtly, from asymmetric segregation problems increasing eccDNA loss to daughter cells (as in *yku70Δ* [58]). We therefore set out to validate key aspects of the eccDNA formation mechanism in unselected populations of young cells. This would reduce age effects because the population is overwhelmingly young and avoid segregation effects because any eccDNA formed can be detected irrespective of whether it ends up in a mother or a daughter cell. Furthermore, we used haploid cells lacking the MEP system and without biotin labelling to rule out any contribution of these experimental tools to the observed phenotype.

To enrich for eccDNA in unselected young populations, we digested genomic DNA samples with ExoV, but even after enrichment the detection of *CUP1*, eccDNA was unreliable and could not be used in quantitative experiments. In contrast, detection of eccDNA from the $P_{GAL1}$-*3HA cup1* allele (S1B Fig) proved robust over multiple experiments and provided a reliable system to test eccDNA formation. Induction of the $P_{GAL1}$ promoter using a 6 hour galactose pulse resulted in formation of eccDNA from the $P_{GAL1}$-*3HA cup1* allele, showing that transcriptional activation of the locus causes eccDNA formation in young cells (Fig 5D upper panel lanes 7, 8). In contrast, formation of eccDNA from this locus in cells lacking Sae2 or Mus81 was not detectable, in accord with our experiments in aged cells (Fig 5D upper panel lanes 9–12). This difference could not be attributed to a defect in transcriptional induction from the $P_{GAL1}$ promoter because induction was similar in all 3 strains (S5 Fig). Curiously, we also observed that ERC levels were reduced in *sae2Δ* cells as well as in *mus81Δ*, albeit to a much lesser extent than for $P_{GAL1}$-*3HA* eccDNA (Fig 5D lower panel). This suggests that the Sae2-dependent pathway can contribute to ERC formation; noncoding RNAs are important in rDNA recombination, and it is possible that ERCs form through both transcriptionally induced and nontranscriptionally induced pathways that are more or less prominent in different media conditions [67]. Nonetheless, this experiment clearly demonstrates that the transcriptionally stimulated formation of eccDNA occurs through a Sae2- and Mus81-dependent DSB processing pathway and that environmentally stimulated eccDNA formation from the *CUP1* locus is not an indirect effect of ageing in general or the experimental details of the MEP system.

Finally, our data suggest that transcriptional activation of the *CUP1* or $P_{GAL1}$-*3HA* allele causes DSB formation but does not exclude other options such as transcription affecting

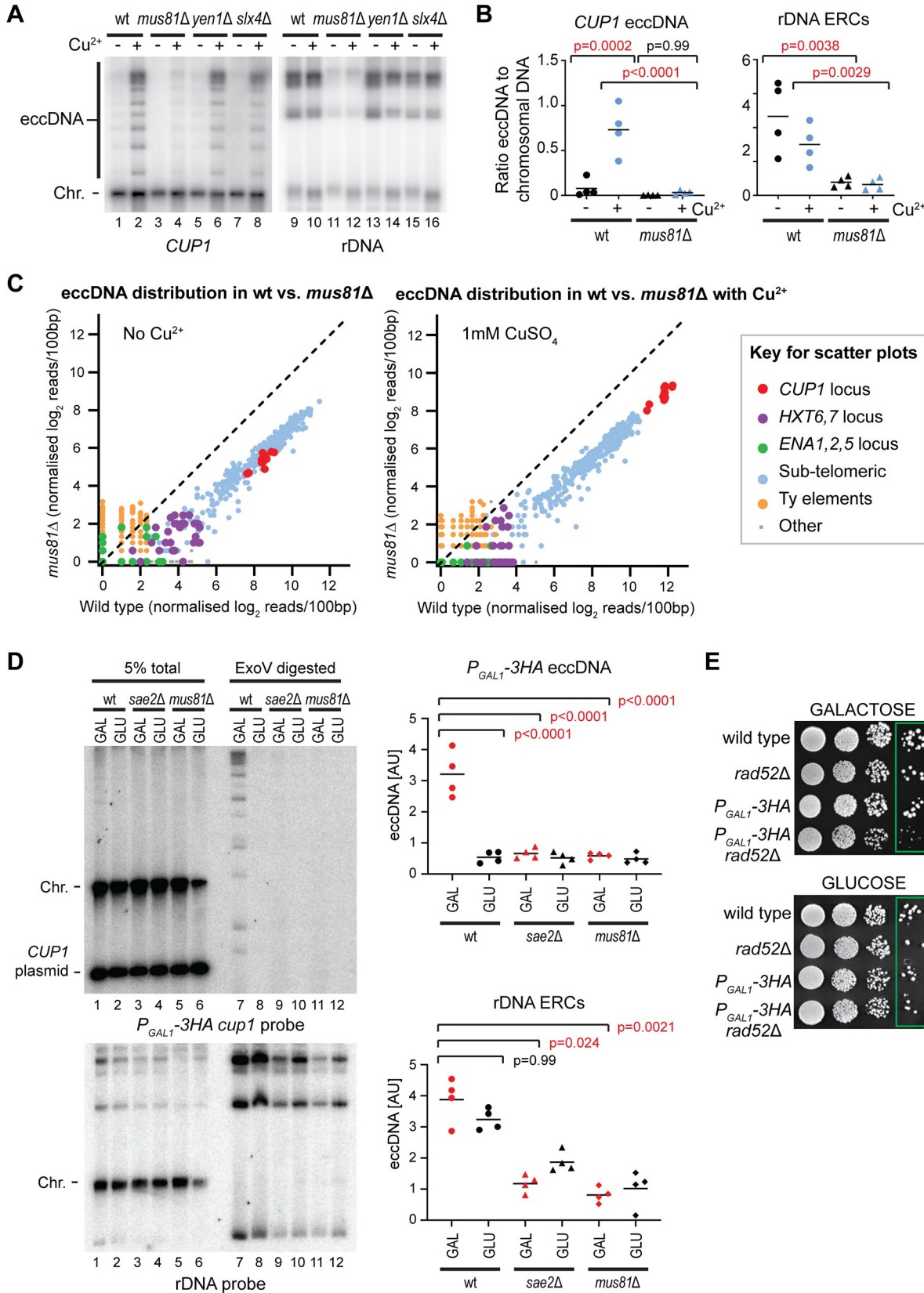

**Fig 5. Mus81 is required for eccDNA formation in old and young cells.** (A) Southern blot analysis of *CUP1* eccDNA and rDNA-derived ERCs in wild-type, *mus81Δ*, *yen1Δ*, and *slx4Δ* cells aged for 24 hours in the presence or absence of 1 mM CuSO₄, performed as in Fig 1B. (B) Quantification of *CUP1* eccDNA and rDNA-derived ERCs in wild-type and *mus81Δ* cells aged for 24 hours in the presence or absence of 1 mM CuSO₄, performed and analysed as in Fig 1B, *n* = 4. (C) REC-seq analysis of *mus81Δ* cells compared to wild type in the absence (left) or presence (right) of 1 mM CuSO₄. Experiment and analysis as in Fig 4D. (D) Southern blot analysis of eccDNA from the 17 copy P_GAL1-*3HA cup1* tandem repeat in non–age-selected BY4741 haploid cell background lacking MEP modifications. P_GAL1-*3HA* wild-type, *sae2Δ*, and *mus81Δ* cells were pregrown on YP Raffinose before a 6 hour induction with 2% galactose or 2% glucose. Genomic DNA was digested with *Xho*I; then 95% of the sample was further digested with ExoV and ExoI; 5% total DNA (lanes 1–6) and 95% ExoV digested material (lanes 7–12) are shown. These cells contain an additional pRS316-*CUP1* plasmid to complement the loss of active chromosomal *CUP1* genes, labelled as *CUP1* plasmid. This plasmid contains an *Xho*I site and is hence linearised by *Xho*I and degraded by ExoV. Signals from same membrane stripped and reprobed for rDNA show ERC species. Abundances of eccDNA and ERCs were compared by one-way ANOVA; *n* = 4 biological replicates; data were log transformed for testing to fulfil the assumptions of a parametric test. (E) Colony formation assay performed on P_GAL1-*3HA* wild-type and *rad52Δ* cells along with BY4741 wild-type and *rad52Δ* controls. Cells were pregrown as above on YP raffinose, then serial dilutions from 10⁴ to 10¹ cells spotted on YPD and YPGal plates, which were grown at 30°C until control cells had formed equivalent sized colonies (2–3 days). The data underlying this figure may be found in S1 Data and S1 Raw Images. eccDNA, extrachromosomal circular DNA; ERC, extrachromosomal ribosomal DNA circle; ExoV, exonuclease V; rDNA, ribosomal DNA; REC-seq, restriction-digested extrachromosomal circular DNA sequencing.

recombination outcome. To provide a more direct insight into DNA damage at this locus, we took advantage of the high efficiency of recombination in the P_GAL1-*3HA cup1* strain. We reasoned that if repair is blocked by deletion of *RAD52* in these cells, then transcriptionally induced DNA damage should cause a prolonged arrest that would manifest in slow colony growth. We therefore plated P_GAL1-*3HA cup1* and P_GAL1-*3HA cup1 rad52Δ* cells on galactose and glucose plates, along with wild-type and *rad52Δ* controls that should not undergo transcriptionally induced DNA damage under these conditions. As predicted, P_GAL1-*3HA cup1 rad52Δ* colonies grew very slowly on galactose relative to P_GAL1-*3HA cup1 RAD52*, whereas growth was equivalent on glucose, and the *rad52Δ* mutation alone had no effect (Fig 5E, particularly the individual colonies highlighted in green boxes). This shows that DNA damage occurs in response to transcription of the P_GAL1-*3HA cup1* allele.

Overall, these data confirm that strong transcriptional induction can cause DNA damage that is processed by Sae2, Mre11, and Mus81 to yield eccDNA.

## Discussion

Here, we have described the reproducible formation of eccDNA from the *CUP1* locus in response to transcriptional activation of the *CUP1* gene. *CUP1* eccDNA accumulates reproducibly to high levels through a combination of frequent formation events and retention of eccDNA in ageing cells.

### A mechanism for the formation of recurrent eccDNA

Formation of *CUP1* eccDNA has a major dependence on Sae2 and on Mre11 nuclease activity. These are processing factors involved in DSB repair by HR (recently reviewed in the work by Oh and Symington [68]). Sae2/Mre11 dependence therefore indicates initiation of recombination from a DSB and argues against eccDNA formation through resolution of re-replication structures by HR or excision of single-stranded DNA (ssDNA) loops by mismatch repair as proposed for microDNA, neither of which processes are expected to be Sae2 dependent [43,69]. Rather, our results are coherent with an intrachromosomal HR process, as proposed for formation of telomeric circles and ERCs [13,70] (Fig 6A).

Excision of eccDNA requires 2 additional strand cleavages beyond the initiating DSB (Fig 6A), and this is not a favoured outcome of intrachromosomal strand invasion. Helicases such as Sgs1 will dissolve D-loops formed by strand invasion to minimise rearrangements during DSB repair [71], and the activity of structure specific endonuclease enzymes capable of

cleaving the nicked Holliday Junctions in a D-loop is tightly restricted to G2/M (reviewed in the work by Kim and Forsburg [72]). Nonetheless, an endonuclease activity is required, and we find Mus81, which has a strong preference for cleaving nicked Holliday Junction intermediates, to be critical for eccDNA formation. This shows that eccDNA must be processed from DSBs present in late G2 or M.

The source of the DSB remains to be determined. Although transcription influences rDNA recombination, this involves a constant rate of DSB formation with a variable outcome [67], whereas the $P_{GAL1}$-3HA system has a variable rate of DNA damage depending on transcription (Fig 5E). Topoisomerase II activity at highly transcribed sites can cause DSBs [73], or replication forks colliding with R-loops can be processed to DSBs. Head-on collision of the replisome with transcribing RNA polymerase II has been shown to cause DNA damage mediated by R-loops in yeast, bacteria, and human cells [74–76]. The general danger to the genome of such collisions is reduced by the THO/TREX complex, which mediates efficient transcription and mRNA export, and by RNase H1 and H2 which degrade R-loops directly [77] but is increased by factors such as Yra1 that stabilise R-loops [78]. Intriguingly, DNA damage does not appear to be a direct effect of collision but involves further processing, because histone mutants have been identified that increase R-loops but not DNA damage, adding a level at which the outcome of DNA damage can be modulated [79]. R-loops are therefore an attractive candidate for initiating eccDNA formation in response to transcription; however, it is not clear that Sae2 would be required for processing R-loop–mediated DSBs because Sae2 is of particular importance in repair of 'dirty' DNA ends that defy direct processing. Such ends are exemplified by chemical damage at ionising radiation-induced breaks or covalently bound proteins such as Spo11 [80,81], so the disruption of eccDNA formation in *sae2Δ* mutants indicates that the transcription-associated DSBs at *CUP1* do not possess 'clean' ends. Alternatively, the importance of Sae2 may lie in the recent observation that Sae2 and Mre11 can remove R-loops via a DSB intermediate that could itself initiate intrachromosomal recombination [82]. Whichever the mechanism, transcriptional stimulation of any recombination process has the potential to bias recombination patterns depending on the gene expression pattern of the cell.

Previous studies have linked ERC formation to DSBs at stalled replication forks [63,83], which appears separable from the transcription-induced Sae2-dependent *CUP1* eccDNA formation mechanism. However, there is clearly crossover between these mechanisms as *CUP1* eccDNA levels in aged non–copper-treated cells are Sae2-independent (Fig 4E), whereas ERCs in log phase cells on YP media are partially Sae2-dependent (Fig 5D). We therefore suggest that both ERCs and *CUP1* eccDNA, and most likely other eccDNA, form both from stalled replication forks and from transcriptionally induced DSBs, with the balance between these mechanisms being variable, dependent on age and environment. In contrast, Mus81 appears to be universally important, which would be consistent with Mus81 resolving the circular intermediate formed by intrachromosomal recombination initiated by either transcriptional or replication-induced DSBs.

eccDNA excision does not repair the initiating chromosomal DSB, and our model invokes a chromosomal deletion matching the excised region (Fig 6A). This can be repaired either by HR with the sister chromatid for DSBs occurring post replication or in a repetitive region by SSA (Fig 6B) [84]. The former pathway can repair the deletion conservatively, whereas the latter necessitates a chromosomal contraction matching or exceeding the excised circle. Where eccDNA is formed from a one-ended DSB at a cleaved replication fork, repair must occur by break induced replication, and as long as the sister chromatids remain in register, this should not result in a loss of chromosomal sequence. Genome resequencing of highly aged cells has not found evidence of repeat contractions suggesting that the conservative modes of recombination are prevalent [85], although some proportion of the *CUP1* contractions we and others have reported may be attributable to SSA [50,51].

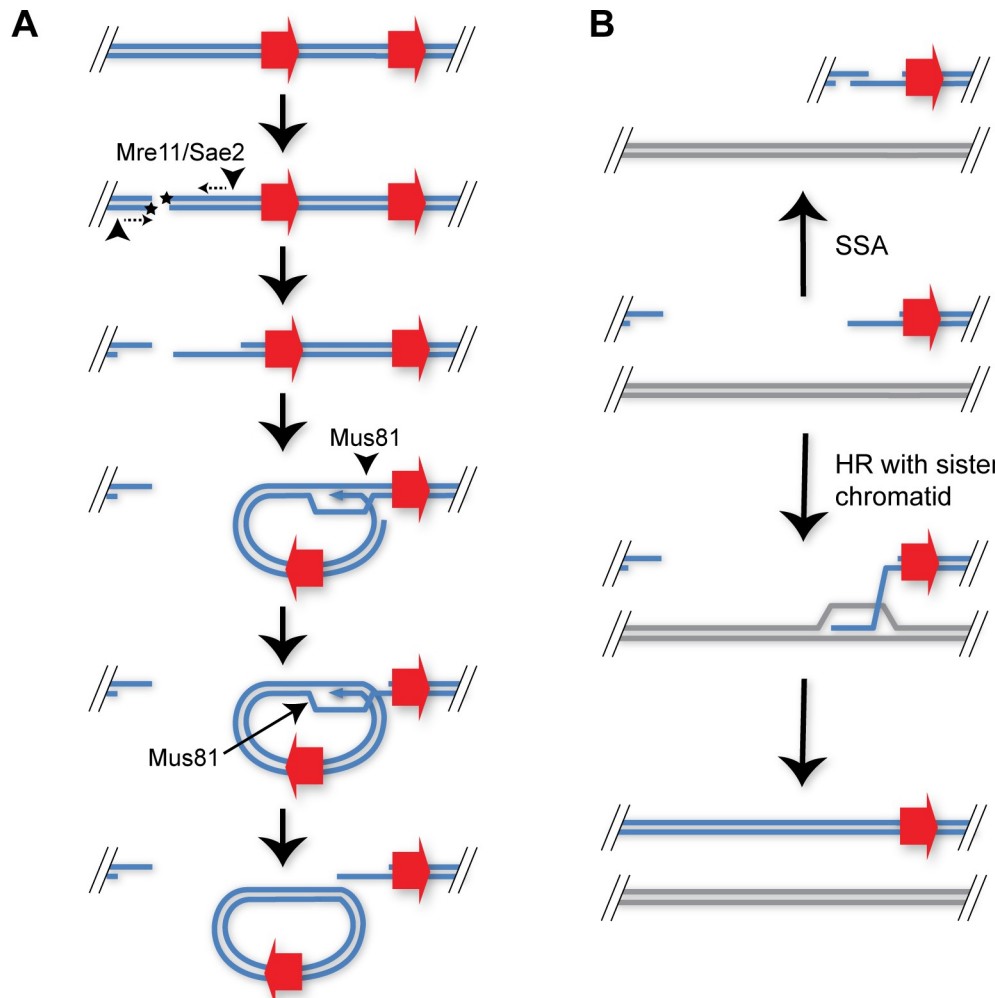

**Fig 6. Proposed mechanisms of eccDNA formation from DNA DSBs and chromosomal repair.** (A) A DSB is initially resected by Mre11 stimulated by Sae2, then invades a homologous sequence in the same chromosome. The D-loop formed is cleaved by Mus81 at 2 separate points, which, after ligation, yields an eccDNA and a DSB with a matching gap. (B) After excision of eccDNA, a partially resected DNA is formed distal to the original DSB site. This can either undergo HR with the sister chromatid resulting in a conservative repair or within a repetitive region such as the *CUP1* locus; this can also be repaired by SSA, resulting in a chromosomal deletion matching the excised circle. DSB, double strand break; eccDNA, extrachromosomal circular DNA; HR, homologous recombination; SSA, single strand annealing.

## eccDNA formation enhances genome plasticity

The de novo formation of eccDNA provides various pathways for adaptation. Even in the absence of reintegration, an eccDNA with an active replication origin can amplify within a population through asymmetric segregation and selection. The presence of a high copy but heterogeneously distributed eccDNA means that all possible gene amplifcation levels are present in the population, so if survival under a particular challenge requires a certain range of copy numbers, some cells within the population should always survive. However, this enhanced adaptibility must be offset against negative effects caused by overexpression of other genes on eccDNA in the absence of the environmental challenge.

In this context, it is interesting that budding yeast effectively follow 2 strategies. By retaining eccDNA in the ageing subpopulation, the majority of young cells are able to grow unimpeded

while the aged cells take on both the increased adaptability, but also the fitness cost, of maintaining eccDNAs. We and others have previously demonstrated an enhanced adaptive capacity within ageing yeast, and the accumulation of additional eccDNA likely represents an additional facet of this [86,87]. Because eccDNA is retained in the mother, it may only benefit the mother rather than any offspring. However, the machinary that retains eccDNA in the mother cell could easily be regulated under environmental stress conditions, and indeed donations of eccDNA from mother to daughter have been observed [32].

Transcriptional stimulation of eccDNA formation biases towards genes that are highly expressed in the current environment, and if inducible, this implies (though does not necessitate) that those genes have evolved to be expressed under the given environment because they are useful and may therefore benefit from copy number amplification. Indeed, the yeast genome is broadly stratified into housekeeping and environmentally responsive genes [88,89], so preferential targeting of DSBs and consequent repair to highly expressed inducible genes is not impossible. The likelihood of copy number variation arising through such repair events also depends on the presence of homologous sequence, and so for genes not already present in multicopy arrays, the distribution of repetitive elements (such as LINE retrotransposons in higher eukaryotes) will significantly impact eccDNA formation. In this scenario, the stimulation of eccDNA formation at inducible genes that are highly expressed in the current environment makes use of information embedded in transcription patterns to direct further adaptive events, pointing to a transcription-driven mechanism conferring increased robustness and/or fitness through enhanced genome plasticity.

## Materials and methods

### Strains and media

Yeast strains were constructed by standard methods and are listed in S1 Table; oligonucleotide sequences are given in S2 Table, and plasmids are given in S3 Table. Most experiments were performed in synthetic complete media SD (2% glucose, 0.67% yeast nitrogen base with ammonium sulphate, complete amino acid mix), to which $CuSO_4$ was added from a 1 M stock solution as required. Where noted, cells were grown in YP media (2% peptone, 1% yeast extract) with sugar at 2% final volume (YPD: YP + glucose, YPGal: YP + galactose), precultures for experiments in YP media were grown in YP with 2% raffinose. In all cases, cells were grown at 30˚C with shaking at 200 rpm. Media components were purchased from Formedium, UK, and all media were sterilised by filtration. For all experiments, cells were brought to log phase through a 2-step culture process: cells were inoculated in 4 ml media and grown for 6 to 8 hours, then diluted in 25 ml or more media and grown for 16 to 18 hours to 0.2 to $0.8 \times 10^7$ cells/ml before further manipulation.

### MEP cell labelling and purification

For biotin labelling, $0.25 \times 10^7$ cells per sample were harvested by centrifugation (15 s at 13,000$g$), washed twice with 125 μl PBS and resuspended in 125 μl of PBS containing approximately 3 mg/ml Biotin-NHS (10538723 ThermoFisher Scientific, UK). Cells were incubated for 30 min on a wheel at room temperature, washed once with 100 μl PBS, and resuspended in 100 μl PBS, then inoculated in required media at $2 \times 10^4$ cells/ml and allowed to recover for 2 hours at 30˚ before addition of 1 μM β-estradiol (E2758 Sigma-Aldrich, UK) and $CuSO_4$ as required. For ageing in YPD or YPGal, the same protocol was followed, but no recovery period was included—as previously noted, this prevents culture saturation without detectable contamination of young cells by 24 to 48 hours [31]. Cells were harvested by repeated

centrifugation for 1 min, 4,600 rpm in 50 ml tubes, and immediately fixed by resuspension in 70% ethanol and stored at −80˚C.

Percoll gradients (1–2 per sample depending on final OD) were formed by vortexing 1.16ml Percoll (P4937 Sigma-Aldrich, UK) with 42 μl 5 M NaCl, 98 μl water in 2 ml tubes and centrifuging 15 min at 15,000$g$, 4˚C. Cells were defrosted, washed with 25 ml cold PBSE, and resuspended in a minimal amount of PBSE and layered on the preformed gradients. Gradients were centrifuged for 4 min at 2,000$g$, then the upper phase and brown layer of cell debris removed and discarded; 1 ml PBSE was added, mixed by inversion, and centrifuged 1 min at 2,000$g$ to pellet the cells, which were then resuspended in 1 ml PBSE (reuniting samples split across 2 gradients). A total of 25 μl Streptavidin magnetic beads were added (1010007 Miltenyi Biotech, Germany), and cells incubated for 15 min on a wheel at room temperature. Meanwhile, 1 LS column per sample (1050236 Miltenyi Biotech, Germany) was equilibrated with cold PBSE in 4˚C room. Cells were loaded on columns and allowed to flow through under gravity, washed with 1 ml cold PBSE and eluted with 1ml PBSE using plunger. Cells were reloaded on the same columns after re-equilibration with approximately 500 μl PBSE, washed and re-eluted, and this process repeated for a total of 3 successive purifications. A total of 50 μl cells were set aside for quality control, while 1 μl 10% Triton X-100 was added to the remainder which were then pelleted by centrifugation and frozen or processed directly for DNA extraction.

For quality control, the 50 μl cells were diluted to 300 μl final volume containing 0.3% triton X-100, 0.3 μl 1mg/ml streptavidin 594 (S11227 ThermoFisher Scientific, UK), 0.6 μl 1 mg/ml WGA-488 (W11261 ThermoFisher Scientific, UK), and 0.1 μg/ml DAPI in PBS. Cells were stained for 15 minutes—overnight at room temperature, washed once with PBS + 0.01% Triton-X100 then resuspended in 7 μl VectaShield (H-1000 Vector Laboratories, Burlingame, California). Purifications routinely yielded 80% to 90% streptavidin positive cells with appropriate bud-scar numbers.

### DNA extraction and Southern blot analysis

For ageing samples, cell pellets were resuspended in 50 μl 0.34 U/ml lyticase (L4025 Sigma-Aldrich, UK) in 1.2 M sorbitol, 50 mM EDTA, 10 mM DTT and incubated at 37˚C for 45 minutes. After centrifugation at 1,000$g$ for 5 minutes, cells were gently resuspended in 80 μl of 0.3% SDS, 50 mM EDTA, 250 μg/ml Proteinase K (3115801001 Sigma-Aldrich, UK), and incubated at 65˚C for 30 minutes. A total of 32 μl 5 M KOAc was added after cooling to room temperature, samples were mixed by flicking, and then chilled on ice for 30 minutes. After 10 minutes centrifugation at 20,000$g$, the supernatant was extracted into a new tube using a cut tip, 125 μl phenol:chloroform (pH 8) was added, and samples were mixed on a wheel for 30 minutes. Samples were centrifuged for 5 minutes at 20,000$g$; the upper phase was extracted using cut tips and precipitated with 250 μl ethanol. Pellets were washed with 70% ethanol, air-dried and left overnight at 4˚C to dissolve in 20 μl TE. After gentle mixing, 10 μl of each sample was digested with 20U *Xho*I or *Eco*RI-HF (R3101 New England Biolabs, Ipswich, Massachusetts) for 3 to 6 hours in 20 μl 1x CutSmart buffer (B7204 New England Biolabs, Ipswich, Massachusetts), 0.2 μl was quantified using PicoGreen DNA (P7589 ThermoFisher Scientific, UK), and equivalent amounts of DNA separated on 25 cm 1% 1x TBE gels overnight at 90 V (120V for gels shown in Fig 1). Gels were washed in 0.25 N HCl for 15 minutes, 0.5 N NaOH for 45 minutes, and twice in 1.5 M NaCl, 0.5 M Tris (pH 7.5) for 20 minutes before being transferred to 20 × 20 cm HyBond N+ membrane in 6x SSC. Membranes were probed using random primed probes (S4 Table) in 10 ml UltraHyb (AM8669 ThermoFisher Scientific, UK) at 42˚C and washed with 0.1x SSC 0.1% SDS at 42˚C or probed in Church Hyb at 65˚C and washed

with 0.5x SSC 0.1% SDS at 65˚C. For probe synthesis, 25 ng template DNA in 38 μl water was denatured for 5 minutes at 95˚C, chilled on ice, then 10 μl 5x labelling buffer (5x NEBuffer 2, 25 μg/ml d(N)$_9$, 165 μM dATP, dGTP, dTTP), 1 μl Klenow exo- (M0212 New England Biolabs, Ipswich, Massachusetts) and varying amounts of α[$^{32}$P]-dCTP added before incubation for 1 to 3 hours at 37˚C, cleaning using a mini Quick Spin DNA column (11814419001 Sigma-Aldrich, UK), and denaturing 5 minutes at 95˚C before adding to hybridisation buffer. During the course of this work, we learnt that controlling the amount of α[$^{32}$P]-dCTP added to the labelling reaction was critical; too high levels resulted in high membrane background irrespective of wash conditions and probe purification. On the activity date, 0.1 to 0.2 μl 3000Ci/mmol α[$^{32}$P]-dCTP (NEG013H Perkin Elmer, Waltham, Massachusetts) was used, an amount that was doubled every 2 weeks past the activity date. Images were obtained by exposure to phosphorimaging screens (GE) and developed using a FLA 7000 phosphorimager (GE, Boston, Massachusetts).

Bands were quantified using ImageQuant version 7.0 (GE) and data analysed using the GraphPad Prism version 6.05. Samples were compared by one-way ANOVA with a Tukey correction for multiple comparisons; where data contravened the assumptions of a parametric test, data sets were log transformed prior to statistical testing.

For DNA samples in Fig 5D, a larger genomic DNA preparation protocol using $20 \times 10^7$ cells per sample was employed as described by Hull and colleagues [50]. Final genomic DNA was dissolved in 50 μl TE at 65˚C for 15 minutes then 4˚C overnight; 10.5 μl 10x CutSmart, 42.5 μl water, and 2 μl XhoI were added and samples digested for 3 hours at 37˚C; 1 μl 10x CutSmart, 1 μl RecBCD [ExoV], 1 μl ExoI, 6 μl 10 mM ATP, and 3.5 μl water were then added to 47.5 μl digested DNA and incubated overnight at 37˚C. Digested DNA was phenol chloroform extracted, ethanol precipitated, and resuspended in 16 μl TE. These samples along with 2.5 μl XhoI digested total DNA were separated and probed as above.

### Northern analysis

Northern analysis was performed as previously described by Cruz and Houseley [90] using probes described in the work by Hull and colleagues [50] and S4 Table.

### Image processing and data analysis

Gel images were quantified using ImageQuant version 7.0 (GE); images for publication were processed using ImageJ version 1.50i by cropping and applying minimal contrast enhancement. Statistical analysis was performed using GraphPad Prism version 7.03.

### REC-seq

All reagents from New England Biolabs, Ipswich, Massachusettsexcept where noted otherwise. A total of 20 μl genomic DNA isolated from aged cells as above was diluted to 155 μl final volume containing 1x CutSmart and 0.75 μl RNase T1 (EN0542 ThermoFisher Scientific, UK), and incubated 15 minutes at room temperature before splitting into 3x 49 μl and 1x 3μl aliquots (the latter was diluted to 16 μl for total DNA isolation). The 49 μl aliquots were digested with 1 μl EagI-HF (R3505), PvuI-HF (R3150), or 0.5μl PvuII-HF (R3151) for 4 hours at 37˚C; 0.5 μl additional enzyme was added to the PvuII digest every hour. For mutant analysis, PvuII was replaced with SmaI (R0141), and this digestion was performed at 25˚C. A total of 6 μl 10 mM ATP, 1 μl 10x CutSmart, 1 μl Exonuclease V, (RecBCD, M0345) and 1 μl Exonuclease I (M0293) was added to each digest and incubated over night at 37˚C before extraction with phenol chloroform and ethanol precipitation in the presence of 1 μl GlycoBlue (AM9516 ThermoFisher Scientific, UK). DNA pellets were dissolved in 45 μl 0.1x TE, then 6 μl 10x CutSmart,

6 μl 10 mM ATP, 1 μl Exonuclease V, 1 μl Exonuclease I, and 1 μl same restriction enzyme as previous day were added and samples again incubated over night at 37˚C before extraction with phenol chloroform and ethanol precipitation. For *Sma*I digestions, the enzyme and CutSmart were added first and digestion performed for 4 hours at 25˚C before addition of other components and shift to 37˚C for overnight incubation. Pellets were dissolved in 33 μl 0.1x TE and the 3 aliquots combined, precipitated again with ethanol and pellets dissolved in 16 μl 0.1x TE.

Exonuclease treated and total DNA samples were incubated at 37˚C for 45 minutes with 2 μl NEBNext DNA fragmentase (M0348) and 2 μl fragmentase buffer; then 5 μl 0.5 M EDTA was added followed by 25 μl water, and samples were purified using 50 μl AMPure beads (A63881 Beckman Coulter, UK) and eluted with 25.5 μl 0.1x TE. 3.5 μl NEBNext Ultra II end prep buffer and 1.5 μl NEBNext Ultra II end prep enzyme (E7103) were mixed in and samples incubated 30 minutes at 20˚C, then 30 minutes at 65˚C. After addition of 1.25 μl 1:25 diluted NEBNext adaptor (E7335), 0.5 μl ligation enhancer and 15 μl NEBNext Ultra II ligation mix (E7103) samples were incubated 15 minutes at 20˚C, then 1.5 μl USER enzyme (E7335) was added and incubated 15 minutes at 37˚C. Samples were cleaned with 44 μl AMPure beads, eluted with 30 μl 0.1x TE, then cleaned again with 27 μl AMPure beads and eluted with 22.5 μl 0.1x TE.

A total of 1.25 μl library was amplified with 0.4 μl each NEBNext index and universal primers (E7335) and 5μl NEBNext Ultra II PCR mix (E7103) in 10 μl total volume using recommended cycling conditions with 8 cycles for total samples, 16 cycles for 48 hour aged samples and 18 cycles for 24 hour samples. These test amplifications were cleaned with 9 μl AMPure beads, eluted with 1.5 μl 0.1x TE and 1 μl run on a Bioanalyzer to determine the optimal cycle number for the final amplification. Library yield was aimed to be 2 pM; the test amplification should be equivalent to a 1:4 dilution of the final library. The remaining library (21 μl) was then amplified in a 50 μl reaction containing 2 μl each NEBNext index and universal primers and 25 μl NEBNext Ultra II PCR mix using the calculated cycle number, cleaned with 45 μl AMPure beads, eluted with 25 μl 0.1x TE, then cleaned again with 22.5 μl AMPure beads and eluted with 10.5 μl 0.1x TE.

## Sequencing and bioinformatics

Libraries were sequenced on a NextSeq500 (Illumina, San Diego, California) in Paired End 75 bp Midoutput mode by the Babraham Institute Sequencing Facility, and basic data processing performed by the Babraham Institute Bioinformatics Facility. After adapter and quality trimming using Trim Galore (version 0.5.0), REC-seq data was mapped to yeast genome R64-2-1 with 2μ sequence J01347.1 included as an additional chromosome using Bowtie2 (version 2.3.2; with more stringent parameters:—no-mixed—score-min L,0,-0.2—no-unal -X 2000). To avoid apparent enrichment of repetitive regions (https://sequencing.qcfail.com/articles/deduplicating-ambiguously-mapped-data-makes-it-look-like-repeats-are-enriched/), the resulting BAM files were subjected to de-duplication based on exact sequence identity of the first 50 bp of both Read 1 and Read 2 using a custom script (https://github.com/FelixKrueger/deduplicate_by_sequence). These de-duplicated by sequence files were then imported into SeqMonk version 1.44.0 for analysis (https://www.bioinformatics.babraham.ac.uk/projects/seqmonk/), where they were additionally de-duplicated based on alignment position (chromosome, start and end). Reads were quantified in 20 bp bins and normalised based on the total read count in each library. For 3x*CUP1*/Δ libraries, counts were performed in 100 bp bins and data from replicate sets compared using the edgeR statistical filter implemented in SeqMonk [91]. Reads from rDNA, mitochondrial, and 2μ regions were excluded except where 2μ was

used for normalisation, because all of these are cleaved by restriction enzymes in REC-seq and give variable signals. Reads from *UBC9* were also excluded because this gene is cleaved in the MEP system to produce a circular DNA in daughter cells, some of which inevitably remain in the purified mother cell fraction [52].

Normalisation of REC-seq data using 2μ read counts in wild-type and mutants: we performed both total DNA-seq and REC-seq for each sample. We then calculated scaling factors ($SF_1$) to transform REC-seq read counts into fractions of nonrepetitive chromosomal read counts as follows:

$SF_1$ = ((2μ reads in total DNA) / (nonrepetitive chromosomal reads in total DNA)) / (2μ reads in REC-seq).

Nonrepetitive reads in total DNA are used, because some repetitive regions produce enough eccDNA to distort total read count (this is visible for subtelomeric regions in S4A Fig).

Simply multiplying REC-seq read counts by $SF_1$ yields unusably low numbers. We therefore divided $SF_1$ for each sample by $SF_1$ for the control condition ($SF_{1c}$, calculated for wild-type without copper) to yield a final scaling factor (SF) for each sample as SF = $SF_1$ / $SF_{1c}$. Because $SF_1$ for the control condition = $SF_{1c}$, SF for the control condition is 1, and as such read counts for the control condition remain unaltered, whereas SF for other conditions provides a minimal linear correction that accounts for the relationship between samples. REC-seq read counts were multiplied by SF. The result of this strategy is that REC-seq reads are normalised to chromosomal DNA, and variation in 2μ levels between mutants does not impact the final normalised REC-seq data (S4 Fig).

All raw REC-seq data has been deposited at GEO under accession number GSE135542.

## Supporting information

**S1 Fig. ExoV digestion validation and schematic of $P_{GAL1}$-*3HA* allele.** (A) Cells were aged in the presence of $CuSO_4$, and genomic DNA purified before being split in 2 aliquots. Both aliquots were digested with *Xho*I, and one was additionally digested with ExoV (aka RecBCD) in the presence of 1 mM ATP. After purification, DNA was separated on a 1% agarose gel and probed for the *CUP1* locus, then stripped and reprobed for *MUT204*, a single copy intergenic region on Chromosome III. (B) Schematic of the *CUP1* locus: detailed view of a single *CUP1* repeat, and map of differences between the wild-type and $P_{GAL1}$-*3HA* allele. In this allele, the *CUP1* ORF and *CUP1* promoter along with part of the annotated ARS810 sequence are removed and replaced by a fused ORF consisting of *3HA* and 6 amino acids of *GAL1* and the $P_{GAL1-10}$ promoter. Restriction sites are shown for orientation, the *Nhe*I site was added during construction between the *CUP1* stop codon and start of the 3' UTR. The data underlying this figure may be found in S1 Raw Images. ExoV, exonuclease V; ORF, open reading frame (TIF)

**S2 Fig. Improved eccDNA sensitivity in REC-seq.** Plots of sequencing reads summed in 200 bp bins across Chromosome XII. (A) Genomic DNA from cells aged for 48 hours in SD media was digested directly with ExoV + ExoI only prior to library preparation. ExoI is included in these reactions to remove short single-stranded products of ExoV digestion. (B) Genomic DNA from cells aged for 48 hours in SD media was split, digested with 3 different enzymes followed by ExoV + ExoI before library preparation. Note the prominent Ty-element peaks in panel B that are not detectable in panel A. The data underlying this figure may be found in S1 Data. eccDNA, extrachromosomal circular DNA; ExoV, exonuclease V; REC-seq, restriction-digested extrachromosomal circular DNA sequencing (TIF)

**S3 Fig. *CUP1* expression is normal in *spt3Δ* cells.** MEP wild-type and *spt3Δ* cells growing in SD media were induced with 1 mM $CuSO_4$ and cells harvested after 0, 4, and 24 hours. Total RNA was separated on a glyoxal gel and probed for *CUP1* ORF, ethidium stained 18S ribosomal RNA is shown as a loading control. Error bars show standard deviation, n = 3. The data underlying this figure may be found in S1 Data and S1 Raw Images. MEP, mother enrichment program
(TIF)

**S4 Fig. Normalisation based on 2μ reads.** (A) Comparison of total DNA sequencing in aged wild-type and *spt3Δ* cells split in 250 bp bins across the genome. Two graphs of the same data are shown, one as a density plot (left), the other highlighting prominent features (right). Most bins representing genomic DNA are on the centre line, significant outliers are labelled—*SPT3* locus is absent in *spt3Δ* cells, 2μ is highly abundant but in this case slightly more prevalent in the wild-type sample. The subtelomeric regions are shifted towards wild type as the high levels of eccDNA produced from these loci adds to the read count even in total DNA. (B) Comparison of eccDNA sequencing from the same 2 samples as in panel A. The normalisation ensures that the ratio of 2μ reads between wild-type and *spt3Δ* is maintained, even though all circles are under-represented in the *spt3Δ* sample, a difference that would not be detected without such normalisation. The data underlying this figure may be found in S1 Data. eccDNA, extrachromosomal circular DNA
(TIF)

**S5 Fig. $P_{GAL1}$-*3HA* expression is minimally affected in *sae2Δ* and *mus81Δ*.** $P_{GAL1}$-*3HA* cells in BY4741 background with no deletion, *sae2Δ* or *mus81Δ* growing in YP raffinose media were induced with 2% galactose and cells harvested after 0, 20, 40, and 60 minutes. Total RNA was separated on a glyoxal gel and probed for *CUP1* ORF, followed by the *ACT1* ORF as a loading control. Error bars show standard deviation, *n* = 2. The data underlying this figure may be found in S1 Data and S1 Raw Images.
(TIF)

**S1 Table. Yeast strains used in this research.**
(XLSX)

**S2 Table. Oligonucleotides used in this research.**
(XLSX)

**S3 Table. Plasmids used in this research.**
(XLSX)

**S4 Table. DNA probes used in this research.**
(XLSX)

**S1 Data. Raw numerical data for figures.**
(ZIP)

**S1 Raw Images. Raw image files for blots.**
(PDF)

## Acknowledgments

We thank Kristina Tabbada and Clare Murnane in the BI Next Generation Sequencing Facility for data generation, Anne Segonds-Pichon and Simon Andrews of the BI Bioinformatics facility for statistical and bioinformatics advice, Birgitte Regenberg and Henrik Moller for helpful discussions, and Tanya Paull and Dan Gottschling for reagents.

## Author Contributions

**Conceptualization:** Ryan M. Hull, Jonathan Houseley.

**Funding acquisition:** Jonathan Houseley.

**Investigation:** Ryan M. Hull, Grazia Pizza, Xabier Vergara, Jonathan Houseley.

**Methodology:** Ryan M. Hull, Michelle King, Felix Krueger.

**Software:** Felix Krueger.

**Supervision:** Jonathan Houseley.

**Visualization:** Ryan M. Hull.

**Writing – original draft:** Jonathan Houseley.

**Writing – review & editing:** Ryan M. Hull.

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
