## [Editor Report · Decision Letter 0]

16 Aug 2019

Dear Jon, 

Good to see you at SMBE. Many thanks for submitting your manuscript entitled "Transcription-induced formation of extrachromosomal DNA during yeast ageing" for consideration as a Research Article by PLOS Biology.

Your manuscript has now been evaluated by the PLOS Biology editorial staff, as well as by an academic editor with relevant expertise, and I'm writing to let you know that we would like to send your submission out for external peer review.

*Please be aware that, due to the voluntary nature of our reviewers and academic editors, manuscripts may be subject to delays during the holiday season. Thank you for your patience.*

Please re-submit your manuscript within two working days, i.e. by Aug 20 2019 11:59PM.

Best wishes,

Roli

Senior Editor

PLOS Biology

---

## [Decision Letter · Decision Letter 1]

13 Sep 2019

Dear Jon,

Thank you very much for submitting your manuscript "Transcription-induced formation of extrachromosomal DNA during yeast ageing" for consideration as a Research Article at PLOS Biology. Your manuscript has been evaluated by the PLOS Biology editors, an Academic Editor with relevant expertise, and by four independent reviewers.

You'll see that all four of the reviewers are positive about your study, but between them they make a number of requests, most of which are textual, but some of which may involve additional analyses and/or extra experimental data. In light of the reviews (below), we are pleased to offer you the opportunity to address the comments from the reviewers in a revised version that we anticipate should not take you very long. We will then assess your revised manuscript and your response to the reviewers' comments and we may consult the reviewers again.

Your revisions should address the specific points made by each reviewer. Please submit a file detailing your responses to the editorial requests and a point-by-point response to all of the reviewers' comments that indicates the changes you have made to the manuscript. In addition to a clean copy of the manuscript, please upload a 'track-changes' version of your manuscript that specifies the edits made. This should be uploaded as a "Related" file type. You should also cite any additional relevant literature that has been published since the original submission and mention any additional citations in your response. 

Before you revise your manuscript, please review the following PLOS policy and formatting requirements checklist PDF: http://journals.plos.org/plosbiology/s/file?id=9411/plos-biology-formatting-checklist.pdf. It is helpful if you format your revision according to our requirements - should your paper subsequently be accepted, this will save time at the acceptance stage.

Please note that as a condition of publication PLOS' data policy (http://journals.plos.org/plosbiology/s/data-availability) requires that you make available all data used to draw the conclusions arrived at in your manuscript. If you have not already done so, you must include any data used in your manuscript either in appropriate repositories, within the body of the manuscript, or as supporting information (N.B. this includes any numerical values that were used to generate graphs, histograms etc.). For an example see here: http://www.plosbiology.org/article/info%3Adoi%2F10.1371%2Fjournal.pbio.1001908#s5.

For manuscripts submitted on or after 1st July 2019, we require the original, uncropped and minimally adjusted images supporting all blot and gel results reported in an article's figures or Supporting Information files. We will require these files before a manuscript can be accepted so please prepare them now, if you have not already uploaded them. Please carefully read our guidelines for how to prepare and upload this data: https://journals.plos.org/plosbiology/s/figures#loc-blot-and-gel-reporting-requirements.

Upon resubmission, the editors assess your revision and assuming the editors and Academic Editor feel that the revised manuscript remains appropriate for the journal, we may send the manuscript for re-review. We aim to consult the same Academic Editor and reviewers for revised manuscripts but may consult others if needed.

We expect to receive your revised manuscript within one month. Please email us (plosbiology@plos.org) to discuss this if you have any questions or concerns, or would like to request an extension. At this stage, your manuscript remains formally under active consideration at our journal; please notify us by email if you do not wish to submit a revision and instead wish to pursue publication elsewhere, so that we may end consideration of the manuscript at PLOS Biology.

When you are ready to submit a revised version of your manuscript, please go to https://www.editorialmanager.com/pbiology/ and log in as an Author. Click the link labelled 'Submissions Needing Revision' where you will find your submission record. 

Best wishes,

Roli

Senior Editor

PLOS Biology

REVIEWERS' COMMENTS:

Reviewer #1:

The manuscript by Hull et al. describes the mechanism by which amplification of the CUP1 gene in yeast occurs through formation of extrachromosomal circles (eccs). The same group had shown in 2017 that growing yeast in the presence of copper leads to CUP1 amplification. Now, using newly developed techniques that allow accurate sequencing and quantification, they show that the amplified units are DNA circles formed by homologous recombination repair of double-strand breaks, apparently induced by transcription. The mechanism is very specific to the region being transcribed. Ecc concentration in aged cells depends upon control of segregation, but apparently their formation does not. Ecc formation in ribosomal DNA occurs by a different mechanism dependent on barriers to replication. It is apparent that a low level of CUP1 ecc is also formed by a different mechanism. Thus, there is clearly more than one mechanism of ecc formation. The double-strand break dependence is based on genetic requirements, Sae2, Mre11 and Mus81. The transcription dependence is based on publications by Fogel, who worked on CUP1 expression and amplification in the 1980s and showed that copper induced transcription of CUP1. It is, however, also supported by experiments that used a GAL promoter that showed circles when grown on galactose and not when grown on glucose. These are the same experiments that demonstrate that selection for copper resistance is not required for the amplification. 

The authors point out that this seems to be a very specific and robust adaptive mechanism, in that amplification occurs only in those genes to be transcribed in a given environment thus allowing increased expression of needed genes. The relationship of mutation to transcription in other systems has been discussed in this context before. The manuscript does not make it plain that this mechanism only operates because there is already tandem chromosomal amplification of the CUP1 locus. Thus the generality of the mechanism appears to be limited, and this should perhaps be mentioned. The work does not reveal how transcription generates double-strand breaks. That will require a different approach. 

Overall this is a delightfully careful and insightful paper that provides substantial mechanistic insights into a specialized mechanism. The figures, including extended data, are necessary for the conclusions being made.

Reviewer #2:

This manuscript investigates extrachromosomal DNA circle formation at the repetitive CUP1 locus in budding yeast. The formation of eccDNA and its unequal segregation during cell division provides a potential adaptive mechanism for gene amplification in response to environmental stress and is commonly seen in cancer. The authors show that eccDNA formation at the CUP1 locus is pronounced in replicatively aged cells, similar to the well-known age-associated accumulation of rDNA circles in yeast. In contrast to rDNA circles, however, CUP1 circles are only observed at high levels if yeast were exposed to environmental copper, and the authors show nicely that this effect is linked to the transcriptional activation of the CUP1 locus. In addition a sequencing strategy is presented that allows recovery and comparison of different eccDNA species. The formation of CUP1 circles is shown to depend on the resection factors Sae2 and Mre11 and the recombinase Mus81, indicating a role of the homologous recombination in circle formation.

The manuscript is clear and well written and the presented experiments are for the most part conclusive. My main concern is the attempted normalization between sequencing results using endogenous 2-micron plasmid ratios as the denominator. Overall, this manuscript presents an intriguing new system that should allow the mechanistic dissection of stress-induced eccDNA formation and is expected to be of substantial interest to the field.

Major points:

1. The normalization approach using 2-micron requires more detailed description in the methods. It is difficult to understand how the 2-micron plasmid ratio was used to normalize the data. On a more general note, I am also not convinced that 2-micron plasmid levels, which are themselves changing (e.g. btw wt and spt3), are a reliable signpost for normalization. How can the authors be certain that apparent changes in the abundance of other eccDNA species are not secondary to changes in 2-micron levels under different experimental conditions? Fluctuations in 2micron levels could in principle explain some of the unexpected differences between samples. I think a simple solution to this problem would be to spike a bacterial plasmid into the sequencing samples and use those easily identifiable sequencing reads as a normalization factor.

2. The sequencing analysis shows sharp boundaries for the CUP1 derived eccDNA but I did not expect the boundaries to lie in the neighboring RSC30 gene. Are there obvious sequence homologies that could explain these boundaries?

3. The authors imply that CUP1 circle formation is the result of DSB-stimulated loop-out of CUP1 copies by homologous recombination. They also draw parallels with rDNA circle production but those circles can form without loss of the chromosomal repeats (e.g. Mansisidor et al, Mol Cell 2018). The presented loop-out model would predict that the chromosomal CUP1 locus should show signs of copy number loss, which should be easily detectable by marker loss or Southern given the relatively small size of the CUP1 locus, in particular when analyzing circle formation in the Gal-inducible system. 

Minor points:

1. I assume the CUP1 plasmid in Figure 5D contains an XhoI site? This could be indicated in the figure legend to avoid confusion why a circular DNA element is sensitive to ExoV.

2. The last section of the discussion feels overly speculative. The potential effect of TATA boxes is certainly worth exploring but the relevance to the current work appears tangential. The presented mechanism implies a need for flanking homologous DNA sequences, which are not expected to be a common enough genomic feature around TATA-driven genes to warrant the broad statements of this paragraph.

3. Please make all code available on GitHub.

Reviewer #3:

The current study uses the CUP1 tandem-repeat locus as a model to examine environmentally stimulated accumulation of extrachromosomal circular DNA (eccDNA) in aging yeast cells, which has potential implications for analogous eecDNA formation in higher eukaryotes. The authors used the mother-enrichment program to obtain a population of aged cells and demonstrated an association between the levels of CUP1 eccDNA accumulation and transcription. Transcription from the endogenous promoter was regulated by copper addition or by galactose addition when a mutant repeat unit was fused to the GAL1 promoter; the effect of transcription was locus specific. As with ERCs (ribosomal DNA circles), the accumulation of eccDNA in aged cells required their asymmetric segregation to mother cells. Importantly, the authors used specific genetic backgrounds to examine the mechanism of eccDNA accumulation. Accumulation in the absence of DNL4 excluded NHEJ and the lack of accumulation in the absence of the initiation of DSB end processing (sae2 or mre11-nd mutants) implicated homologous recombination (HR). A requirement for Mus81 (but not Yen1 of Slx4) is consistent with an HR-based mechanism that requires the processing of nick-containing branched intermediates. Interestingly, eccDNA accumulation has variable genetic requiremnents relative to ERC accumulation, which is related to the strong replication block in rDNA. The manuscript is well-written, the data are robust, and the results are important and of broad general interest. Below are comments that should be addressed when revising the manuscript.

1. If the frequency of DSB formation is high enough to cause a growth defect in a rad52 background (which is not a direct demonstration of DSBs), then it should be possible to physically “see” the broken chromosome on a CHEF gel. If the authors haven’t tried this, they should. eccDNAs may be only a very rare outcome of DSB repair, but this doesn’t make them any less interesting.

2. The authors need to provide more info about the pGAL1-3HA construct (consulting the reference cited was not much help). When the CUP1 coding sequence is deleted, how much of the repeat unit remains? Presumably most of it, but this should be stated.

3. Following DSB induction in yeast, repair is delayed but not prevented in the absence of MRX/Sae2. Does this imply the presence of damage at the ends of spontaneous DSBs? The authors should comment on this.

4. There have been MANY studies of transcription-associated recombination (TAR) in yeast, mostly from the Aguilera lab, most of which results from the persistence of R-loops during head-on conflicts between replication and transcription. The Cimprich lab has shown a similar connection in mammalian cells. These studies should be more prominently highlighted as they may well be directly relevant to eccDNA formation.

5. The production of eccDNAs by the current mechanism requires pre-existing direct repeats, which are very rare in the yeast genome. In higher eukaryotes flanking repetitive elements (e.g. LINEs) may be the drivers.

6. The papers the authors cited for TopIIbeta-mediated DSB formation, which is specifically related to transcription initiation in mammalian cells, may not be relevant in yeast.

7. Please note that Tom Petes’ lab has also shown that transcription stimulates recombination (in the form of CNV) at the CUP1 locus (Genetics 2017).

Reviewer #4:

This ms. explores an important problem concerning the generation of eccDNA in budding yeast. They have modified the methods used to measure eccDNA and have made several interesting observations on the copper gene multiple tandem array. The work is very well presented (for the most part) and will be an important contribution to the literature.

Please address the questions at the end of each section from Pg. 4: The genome-wide profile of eccDNA was more affected by copper treatment in these cells, likely due to greater copper stress, and many eccDNAs accumulated to lower levels (Figure 2G). However, edgeR analysis (Where is the edgeR data?) 

of differential accumulation across 3 biological replicates revealed that the significant regions most over-represented in the copper-treated sample all derive from the CUP1 locus. Additionally, Ty-element eccDNA was under-accumulated in copper treated cells (was this significant or not?),

while sub-telomeric eccDNA was over-accumulated though by a very small amount (again, was this significant? - what is the metric here “a very small amount”?).

In Figure 5A and B: The figure legend is little confusing. Are the two blots exactly the same experimental conditions and the data from each is then represented in the right side of scatter plot of 5B? If so, consolidate and leave out the left side of 5B (the same can be done in other figures if applicable).

In Figure 6 and in the discussion, it is not at all clear why propose a gap in the circle at the end of the process? Trimming and ligation, as shown in earlier steps can be invoked without an obligatory gap.

---

## [Editor Report · Decision Letter 2]

18 Oct 2019

Dear Jon,

Many thanks for submitting your revised Research Article entitled "Transcription-induced formation of extrachromosomal DNA during yeast ageing" for publication in PLOS Biology. The Academic Editor and I have now assessed your revisions, and we're delighted to let you know that we're now editorially satisfied with your manuscript.

However before we can formally accept your paper and consider it "in press", we also need to ensure that your article conforms to our guidelines. A member of our team will be in touch shortly with a set of requests. As we can't proceed until these requirements are met, your swift response will help prevent delays to publication. IMPORTANT: Many thanks for providing the underlying data so conscientiously; please could you also cite this Supplementary Data file clearly in the relevant main and supplementary Figure legends, e.g. "The data underlying this figure may be found in S5_Data"?

Please note that you may have the opportunity to make the peer review history publicly available. The record will include editor decision letters (with reviews) and your responses to reviewer comments. If eligible, we will contact you to opt in or out.

Early Version: Please note that an uncorrected proof of your manuscript will be published online ahead of the final version, unless you opted out when submitting your manuscript. If, for any reason, you do not want an earlier version of your manuscript published online, uncheck the box. Should you, your institution's press office or the journal office choose to press release your paper, you will automatically be opted out of early publication. We ask that you notify us as soon as possible if you or your institution is planning to press release the article.

Best wishes,

Roli

Senior Editor

PLOS Biology

For manuscripts submitted on or after 1st July 2019, we require the original, uncropped and minimally adjusted images supporting all blot and gel results reported in an article's figures or Supporting Information files. We will require these files before a manuscript can be accepted so please prepare them now, if you have not already uploaded them. Please carefully read our guidelines for how to prepare and upload this data: https://journals.plos.org/plosbiology/s/figures#loc-blot-and-gel-reporting-requirements.

---

## [Editor Report · Decision Letter 3]

31 Oct 2019

Dear Dr Houseley,

On behalf of my colleagues and the Academic Editor, David Gresham, I am pleased to inform you that we will be delighted to publish your Research Article in PLOS Biology. 

Early Version

PRESS 

Kind regards,

Alice Musson

Publication Assistant, 

PLOS Biology

on behalf of

Roland Roberts,

Senior Editor

PLOS Biology